# LEARNING FROM THE FUTURE: IMPROVE LONG-TERM MESH-BASED SIMULATION WITH FORESIGHT

## ABSTRACT

This paper studies the problem of learning mesh-based physical simulations, a crucial task with applications in fluid mechanics and aerodynamics. Recent works typically utilize graph neural networks to produce next-time states on irregular meshes by modeling interacting dynamics, and then adopt iterative rollouts for the whole trajectories. However, these methods cannot achieve satisfactory performance in long-term predictions due to the failure of capturing long-term dependency and potential error accumulation. To tackle this, we introduce a new future-to-present learning perspective, and further develop a simple yet effective approach named Foresight And InteRpolation (FAIR) for long-term mesh-based simulations. The main idea of FAIR is to first learn a graph ODE model for coarse long-term predictions and then refine short-term predictions via interpolation. Specifically, FAIR employs a continuous graph ODE model that incorporates past states into the evolution of interacting node representations, which is capable of learning coarse long-term trajectories under a multi-task learning framework. Then, we leverage a channel aggregation strategy to summarize the trajectories for refined short-term predictions, which can be illustrated using an interpolation process. Through pyramid-like alternative propagation between the foresight step and refinement step, FAIR can generate accurate long-term trajectories, achieving an error reduction of up to 25.4% on benchmark datasets. Extensive ablation studies and visualization further validate the superiority of the proposed FAIR.

## 1 INTRODUCTION

Physics simulations are of paramount importance for understanding fundamental principles in various domains, including mechanics (Ren & Tang, 2010; Wang et al., 2022), electromagnetics (Pardo et al., 2007), biology (Wang & Wu, 2021) and acoustics (Marburg & Nolte, 2008). The majority of studies utilize mesh-based finite element systems to describe complicated physics by simulating the interactions of mesh points. To achieve the optimal use of resource budgets for unstructured surfaces, they usually allocate greater resolution to regions of interest where more accurate analysis is expected, resulting in complicated irregular mesh structures (Guskov et al., 2002; Liu et al., 2022a; Dong et al., 2023; Liang et al., 2022). Traditional numerical solvers usually require a heavy computational burden, and thus efficient data-driven simulators have drawn ever-lasting interest recently.

With the rapid development of deep learning techniques, several data-driven simulators have been recently proposed to learn numerical simulations on structured grids (Fotiadis et al., 2020; Kim et al., 2019; Tompson et al., 2017; Rao et al., 2023; Cao et al., 2023). To adapt to irregular meshes, graph machine learning-based approaches have received more attention gradually (Pfaff et al., 2021; Xu et al., 2021; Shao et al., 2022; Sanchez-Gonzalez et al., 2020). The majority of them first construct a geometric graph where mesh points are considered as nodes and utilize graph neural networks (GNNs)to model the interacting dynamics in physical systems. In particular, they follow the paradigm of message passing (Kipf & Welling, 2017; Xu et al., 2019; Wu et al., 2019; Alet et al., 2019; Yu et al., 2018; Yang et al., 2022; Zhang et al., 2021) , which aggregates edge information from the neighbors of each node to update the node representation in a progressive fashion.

In reality, long-term forecasting (Nie et al., 2022; Zhou et al., 2022a; Zhao et al., 2020; Lan et al., 2022; Wu et al., 2021) is a practical yet challenging scenario for physics simulations. Existing methods (Pfaff et al., 2021; Shao et al., 2022; Cao et al., 2023) usually rely on

an autoregressive strategy, which utilizes the current states for the next-time predictions and then feeds them back as input in an iterative manner. However, these one-step predictors often struggle to capture the long-term system dynamics, which could be governed by underlying partial differential equations (PDEs) (Vadeboncoeur et al., 2023). Additionally, they are prone to accumulating errors over iterative rollouts, which degrades the performance of their long-term predictions. Given these accumulated errors, it is highly anticipated to include future states into the prediction procedure to provide a foresight of systems, which not only enhances long-term predictions, but also infers extra knowledge connected with underlying PDEs to refine the short-term predictions.

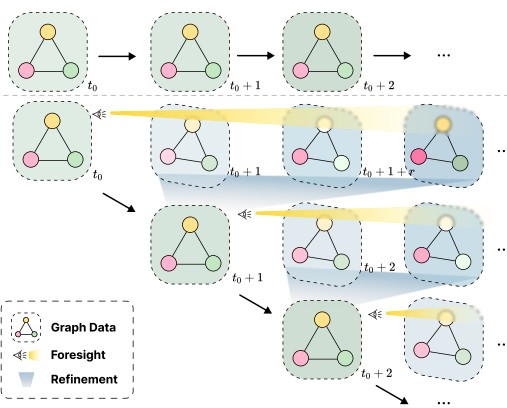

Figure 1: Previous approaches adopt the current states for next-time predictions while our FAIR includes alternative foresight and refinement.

Towards this end, we provide a new perspective that learns from the future for present predictions (see Figure 1), and propose a simple yet effective approach named Foresight And InteRpolation (FAIR) for long-term mesh-based simulations. In particular, FAIR takes a two-stage learning paradigm which first generates coarse long-term predictions using a continuous graph ordinary differential equation (ODE) model, and then refines these into accurate short-term predictions through interpolation. In the first stage, we extend neural ODEs (Chen et al., 2018; Norcliffe et al., 2021) into graph ODE twins to provide coarse foresight, which models the evolution of both mesh node and edge representations using the neighborhood aggregation mechanism. To enhance the capacity to capture non-linear complex patterns, we incorporate historical states to augment the current embedding in ODEs. Our graph ODE model has the flexibility to generate various predictions of different steps ahead, which is optimized for a multi-task learning framework (Momma et al., 2022). In summary, our model can not only accord with the continuous nature of real-world systems, but also capture long-term dependencies with limited error accumulation. In the second stage, we employ a channel aggregation strategy to summarize the future predictions and current observations for interpolation, which is followed by further neighborhood aggregation in the observation space to refine the short-term predictions. More importantly, through the alternative foresight step and refinement step, our FAIR can provide refined long-term trajectories. We validate the effectiveness of FAIR via extensive experiments on four benchmark datasets, and our FAIR achieves significant improvements over various baseline models. In particular, our proposed FAIR achieves an error reduction up to 25.4% on *CylinderFlow* compared with the best baseline.

In summary, the contributions of this paper are three-fold: (1) *Innovative Perspective*. We introduce a new future-to-present perspective for mesh-based simulations, which can effectively capture long-term dynamics with minimal error accumulation. (2) *Novel Methodology*. Our two-stage FAIR first produces coarse long-term predictions using a continuous graph ODE model, and then refines these into accurate short-term predictions through interpolation. These two steps are conducted alternatively for refined long-term predictions. (3) *High Performance*. Extensive experiments on four benchmark datasets demonstrate the superiority of FAIR over existing approaches.

## 2 RELATED WORK

**Learning-based Physics Simulations.** Physics simulations can be advantageous in science when model parameters or boundary conditions are insufficient. Due to their great efficacy, learning-based physics simulations are gaining popularity in a variety of domains such as computational fluid dynamics (Yang et al., 2017; Guo et al., 2016; Qiao et al., 2020; Maulik et al., 2021; Suh et al., 2022; Takamoto et al., 2023). Initially, convolutional neural network-based methods are often built to learn from regular grids (Peng et al., 2020). Recently, MeshGraphNet (Pfaff et al., 2021) makes an attempt to incorporate GNNs into learning mesh-based physics simulations on irregular meshes, followed by several extensions (Shao et al., 2022; Sanchez-Gonzalez et al., 2020; Cao et al., 2023). However, these algorithms often generate next-step predictions using current states, which fails to make accurate long-term predictions. Han et al. (2022) adopts an encoder-decoder structure

to compress data for long sequence modeling with Transformer. In contrast, our proposed FAIR provides a future-to-present perspective to capture long-term dynamics using neural ODE-based models.

**Long-term Forecasting.** Based on historical observations, long-term forecasting (Nie et al., 2022; Zhao et al., 2020; Lan et al., 2022; Wu et al., 2021; Dendorfer et al., 2022; Malhan & Mittal, 2022; Zhou et al., 2022b; Mangalam et al., 2021) aims to make predictions for a long horizon with various applications with weather forecasting (Bi et al., 2023; Zhang et al., 2023) and economic analysis (Chudik et al., 2021). A range of Transformer-based architectures (Zhou et al., 2022a;b; Liu et al., 2022b; Tang & Matteson, 2021) have been introduced for long-term forecasting, which can get rid of gradient vanishing and exploding in recurrent neural networks (RNNs) (Salinas et al., 2020). These approaches usually focus on modeling single-agent systems. In contrast, we target a less-explored and challenging problem of long-term interacting dynamics forecasting (Kofinas et al., 2021) and propose an approach FAIR, which learns coarse long-term trajectories using a graph ODE model to provide future information for short-term refinement. which learns coarse long-term trajectories using a graph ODE model to provide future information for short-term refinement.

## 3 PRELIMINARIES

### 3.1 PROBLEM FORMULATION

We aim to learn a neural simulator that uses neural operations to approximate the ground-truth physics dynamics on irregular meshes, which are usually driven by underlying PDEs. The dynamic system with both spatial correlations can be characterized using a mesh graph $G = (\mathcal{V}, \mathcal{E})$ with a set of mesh nodes $\mathcal{V}$ and an edge set $\mathcal{E}$. Given current states of mesh points, *i.e.*, $\boldsymbol{X}^{t_0} \in \mathbb{R}^{N \times F}$ and (optionally) a series of historical states, *i.e.*, $\{\boldsymbol{X}^{t_0-1}, \cdots, \boldsymbol{X}^{t_0-\tau}\}$, the objective is to predict the future trajectories for all nodes $\boldsymbol{X}^t \in \mathbb{R}^{N \times F}$ ($t_0 < t \leq t_0 + T$), where $F$ is the attribute dimension and $N$ is the number of mesh nodes. We use prediction errors w.r.t. the ground truth to evaluate the performance.

### 3.2 GRAPH NEURAL NETWORKS (GNNS)

GNNs are a class of neural networks that operate directly on graph-structured data (Wei et al., 2022). They have been extensively studied for approximating pairwise interactions in mesh-based physical systems (Pfaff et al., 2021; Shao et al., 2022; Sanchez-Gonzalez et al., 2020). GNNs usually follow the paradigm of message passing (Kipf & Welling, 2017; Veličković et al., 2018; Xu et al., 2019), where edge information is updated by its connected nodes and then neighborhood information is aggregated to update the node representations. Through this, GNNs can capture the complex interaction among mesh points, which reveals how the system changes from time step $t_0$ to time step $t_0 + 1$ (Cao et al., 2023; Yıldız et al., 2022; Look et al., 2023).

### 3.3 NEURAL ORDINARY DIFFERENTIAL EQUATIONS (ODES)

A complex physical system can be described by a series of coupled nonlinear ordinary differential equations (Shao et al., 2022; Han et al., 2022):

$$\frac{d\boldsymbol{h}_i^t}{dt} = \Phi(\boldsymbol{h}_1^t, \boldsymbol{h}_2^t, \cdots, \boldsymbol{h}_N^t), \tag{1}$$

where $\boldsymbol{h}_i^t$ is the state for object $i$ at time step $t$ and $\Phi(\cdot)$ is a function for capturing the interaction among objects, which can be a neural network automatically learned from data (Yoon et al., 2022; Huang et al., 2020; Chen et al., 2018; Huang et al., 2021). Given the initial states $\boldsymbol{h}_1^{t_0}, \boldsymbol{h}_2^{t_0} \cdots \boldsymbol{h}_N^0$ for all objects, the latent states of trajectories at arbitrary time steps can be calculated with a black-box ODE solver as follows:

$$\boldsymbol{h}_i^t = \boldsymbol{h}_i^{t_0} + \int_{s=t_0}^t \Phi\left(\boldsymbol{h}_1^s, \boldsymbol{h}_2^s, \cdots, \boldsymbol{h}_N^s\right) ds. \tag{2}$$

We model mesh-based physical system dynamics using neural ODEs in the latent space, with GNN as the ODE function $\Phi$ to model the continuous interaction among mesh points. The latent initial states $\boldsymbol{h}_1^{t_0}, \boldsymbol{h}_2^{t_0}, \cdots, \boldsymbol{h}_N^{t_0}$ are computed via an encoder and the decoder recovers the whole trajectory $\boldsymbol{X}^t$ ($t_0 < t \leq T$) based on the latent states at each time step.

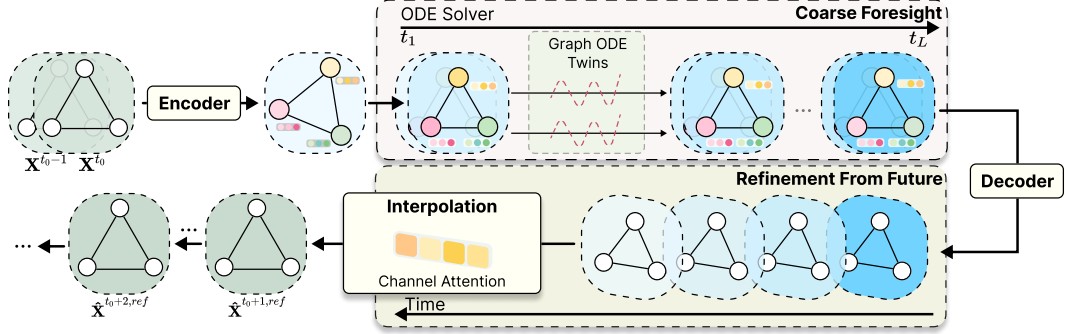

Figure 2: An overview of the proposed FAIR. FAIR adopts an MPNN-based encoder to generate node representations, which are fed into graph ODE to generate predictions at different timestamps. These predictions are aggregated with channel attention to refine the short-term predictions. These foresight and refinement steps are conducted alternatively for accurate long-term predictions.

## 4 THE PROPOSED FAIR

In this paper, we study the problem of long-term mesh-based simulations and introduce a new approach named FAIR from a future-to-present perspective. Existing methods (Pfaff et al., 2021; Xu et al., 2021; Shao et al., 2022) usually utilize the current states for next-time predictions, followed by iterative rollouts to predict whole trajectories while our FAIR takes a future-to-present perspective, allowing the model to maintain foresight throughout the evolution. Specifically, FAIR incorporates a continuous graph ODE model that is enhanced with past states, which generates coarse long-term trajectories under a multi-task learning framework. To improve the accuracy of our short-term predictions, we employ a channel aggregation strategy, which refines the trajectory through an interpolation process. Finally, we adopt pyramid-like propagation for the whole trajectories. An overview of FAIR can be found in Figure 2 and we will elaborate on the details as follows.

### 4.1 COARSE FORESIGHT WITH GRAPH ODE TWINS

The key insight of our FAIR is to learn from long-term trajectories. As a preliminary step, it is crucial to generate high-quality long-term trajectories based on historical predictions. Instead of using inefficient iterative rollouts (Pfaff et al., 2021; Cao et al., 2023), we follow the idea of neural ODEs (Chen et al., 2018) to model the continuous evolution of both mesh nodes and edges within dynamic systems. To learn the evolution between mesh points, we extend neural ODEs into graph ODE twins, which employ a neighborhood aggregation mechanism to update the representations of both nodes and edges, which are flexible to produce outputs at any given timestamp. In particular, FAIR leverages an encoder-ODE-decoder architecture where both the encoder and decoder components are built upon message passing neural networks (MPNNs). The effectiveness of the graph ODE twins is further enhanced by augmenting latent states with historical data, thereby improving the capability to capture evolving patterns under potential noise.

**MPNN-based Encoder.** To begin with, we first generate state representations for mesh nodes and their associated edge using a message passing mechanism (Kipf & Welling, 2017). Specifically, both node and edge embeddings are initialized using feed-forward networks (FFNs) as follows:

$$\boldsymbol{v}_i^{(0)} = f^n\left(\boldsymbol{x}_i\right), \boldsymbol{e}_{ij}^{(0)} = f^e\left(\boldsymbol{p}_i - \boldsymbol{p}_j\right), \tag{3}$$

where $\boldsymbol{x}_i$ and $\boldsymbol{p}_i$ are the feature and position vectors of node $i$, respectively. $f^n(\cdot)$ and $f^e(\cdot)$ are implemented by two FNNs for nodes and edges, respectively. Then, we stack a range of MPNN layers to learn semantics from geometric graphs in an iterative manner. The updating rule for each node $i$ can be summarized as follows:

$$\boldsymbol{v}_i^{(l+1)} = \psi^n(\boldsymbol{v}_i^{(l)}, \sum_{j \in \mathcal{N}(i)} \boldsymbol{e}_{ij}^{(l)}), \boldsymbol{e}_{ij}^{(l+1)} = \psi^e(\boldsymbol{v}_i^{(l)}, \boldsymbol{v}_j^{(l)}, \boldsymbol{e}_{ij}^{(l)}), \tag{4}$$

where $\boldsymbol{v}_i^{(l)}$ and $\boldsymbol{e}_{ij}^{(l)}$ are the node and edge embeddings at the layer $l$, respectively. $\mathcal{N}(i)$ denotes the neighbors of node $i$. $\psi^e$ and $\psi^n$ are two FFNs for feature transformations. After stacking $L$ layers,

we can generate discriminative node end edges representations for the current timestamp $t_0$, *i.e.*, $\boldsymbol{v}_i^{t_0} = \boldsymbol{v}_i^{(L)}$ and $\boldsymbol{e}_{ij}^{t_0} = \boldsymbol{e}_{ij}^{(L)}$ for the subsequent generative model.

**Graph ODE.** Neural ODEs are commonly used to model dynamical systems with continuous evolution, which can output flexible predictions at any given timestamps. While previous approaches often integrate neighborhood information into ODEs to model interacting dynamics (Huang et al., 2020; Gupta et al., 2022), they typically fall short in explicitly capturing the evolving dynamics of edges. To address this gap, we introduce a propagator named graph ODE twins, which adopts separate neural ODEs to model the evolution of both nodes and edges. Furthermore, we notice that data-driven prediction models (Huang et al., 2020; 2021) often struggle to accurately deduce continuous evolution solely based on the current state. To mitigate this, we incorporate historical states as supplementary data in our graph ODE model. In practice, we use them to augment the current embeddings in ODEs, thereby enhancing the capacity to capture the evolving dynamics as well. In formulation, we generate augmented embeddings at the timestamp $t$ as:

$$\bar{\boldsymbol{v}}_i^t = \left[ \begin{array}{c} \boldsymbol{v}_i^t \\ \boldsymbol{v}_i^{t-1} \end{array} \right], \bar{\boldsymbol{e}}_{ij}^t = \left[ \begin{array}{c} \boldsymbol{e}_{ij}^t \\ \boldsymbol{e}_{ij}^{t-1} \end{array} \right], \tag{5}$$

where $\boldsymbol{v}_i^{t-1}$ and $\boldsymbol{e}_{ij}^{t-1}$ are from the last timestamps. Then, we model the evolution using both augmented node embeddings and edge embeddings based on the following ODEs:

$$\frac{d\boldsymbol{v}_i^t}{dt} = \Phi^n(\bar{\boldsymbol{v}}_i^t, \sum_{j \in \mathcal{N}(i)} \bar{\boldsymbol{e}}_{ij}), \frac{d\boldsymbol{e}_{ij}^t}{dt} = \Phi^e(\bar{\boldsymbol{e}}_{ij}^t, \bar{\boldsymbol{v}}_i^t, \bar{\boldsymbol{v}}_j^t), \tag{6}$$

where $\Phi^n$ and $\Phi^e$ are implemented by two FFNs. Through a standard ODE solver, we are able to output the hidden embeddings for the future timestamps ranging from $t_1$ to $t_L$ at one time with the step size $r$ and the number of predictions $L$ (*i.e.*, $t_l = t_0 + rl - r + 1$). Different $r$ indicates a different horizon of predictions. Our graph ODE model is a special case of delay differential equations (DDEs) (Balachandran et al., 2009), *i.e.*, $\frac{d\boldsymbol{v}^t}{dt} = \phi(\boldsymbol{v}^t, \boldsymbol{v}^{t-\tau}, t)$, which has been shown to have an improved capacity for capturing non-linear dynamics (Zhu et al., 2021). We further provide a theorem to show that our graph ODE has a unique absolutely continuous solution. To begin, denote $\boldsymbol{y}^t = (\boldsymbol{v}_1^t, \ldots, \boldsymbol{v}_N^t, \boldsymbol{e}_{12}^t, \ldots, \boldsymbol{e}_{N-1,N}^t)$, and then our system can be represented as:

$$\begin{cases} \dfrac{d\boldsymbol{y}^t}{dt} = \Phi(\boldsymbol{y}^t, \boldsymbol{y}^{t-1}), & t \in [t_0, t_0 + T] \\ \boldsymbol{y}(t) = \boldsymbol{y}(t_0 - 1), & t \in [t_0 - 1, t_0), \end{cases} \tag{7}$$

where we add the definition of $\boldsymbol{y}$ in the interval $[t_0 - 1, t_0]$ to make sure our system is well-defined.

**Lemma 4.1.** *Suppose we have an FFN $\Phi$ with all absolute values of weights and biases bounded by satisfy $M, B$ respectively. Besides ReLU is adopted as the activation function. Then, our Eqn. 7 has a unique absolutely continuous solution.*

The proof has been shown in Appendix A. Through our analysis, we show that the future trajectories are predictable based on the historical states, a crucial property in system dynamics modeling. Furthermore, our predictions can preserve the continuity embedded in mesh-based simulations.

**MPNN-based Decoder.** In the end, we adopt a decoder $\psi_{dec}(\cdot)$ to generate the predictions at any given timestamps as follows:

$$\hat{\boldsymbol{x}}_i^t = \psi_{dec}(\{\boldsymbol{v}_i^t\}_{i \in \mathcal{V}}, \{\boldsymbol{p}_{ij}\}_{(i,j) \in \mathcal{E}}), \tag{8}$$

where $\boldsymbol{p}_{ij} = \boldsymbol{p}_i - \boldsymbol{p}_j$ is reused to provide position information. The architecture of the decoder is the same as the MPNN in the encoder to ensure effective neighborhood learning. To train our graph ODE twins, we minimize the mean square error for different timestamps as follows:

$$\mathcal{L}_{ode} = \sum_{l=1}^{L} \sum_{i=1}^{N} ||\boldsymbol{x}_i^{t_l} - \hat{\boldsymbol{x}}_i^{t_l}||_2, \tag{9}$$

where each timestamp $t_l$ corresponds to a different $t_l$-step ahead prediction task.

**Comparison with One-step Predictors.** Current one-step predictors generate the state for the next-time states (Pfaff et al., 2021; Shao et al., 2022; Cao et al., 2023) and then proceed in an autoregressive manner for the entire trajectories. In contrast, our graph ODE twins have two strengths

as follows. Firstly, our approach is capable of capturing the continuous interactive dynamics that naturally occur in the mesh-based physical system. Secondly, our approach is optimized under the framework of multi-task learning. In particular, we generate predictions with different prediction lengths, each of which corresponds to a task. This strategy enables the model to learn long-term dependency with limited error accumulation, thereby enhancing the learning process.

## 4.2 REFINEMENT WITH INTERPOLATION

While our graph ODE twins model is effective, it has the potential to underfit long-term trajectories in the multi-task learning framework. To address this issue, we introduce a refinement module, which uses the coarse long-term trajectories to improve the accuracy of short-term predictions, *i.e.*, $\hat{\boldsymbol{x}}_i^{t_1}$. Given that both future states beyond the target one, *i.e.*, $\{\hat{\boldsymbol{x}}_i^{t_l}\}_{l=2}^L$ and current states, *i.e.*, $\hat{\boldsymbol{x}}_i^{t_0}$ are both available, this refinement process can be viewed as a form of interpolation.

To be specific, we introduce a set of learnable parameters to serve as channel attention, which are the weights for interpolation. Moreover, the message passing procedure is performed in the observation space rather than the embedding space to learn the offset with enhanced efficiency. In formulation, we define the learnable weights as $\boldsymbol{w}^l \in \mathbb{R}^F$, and the aggregated observation $\boldsymbol{z}_i^{t_1}$ for node $i$ can be written as:

$$\boldsymbol{z}_i^{t_1} = \boldsymbol{x}_i^{t_0} \odot \boldsymbol{w}^0 + \sum_{l=1}^L \hat{\boldsymbol{x}}_i^{t_l} \odot \boldsymbol{w}^l, \tag{10}$$

where target states $\hat{\boldsymbol{x}}_i^{t_1}$ is also involved for a more comprehensive offset mining and $\odot$ denotes element-wise product of two vectors. Compared to standard interpolation, our approach facilitates information exchange across different channels, thereby enhancing the capacity to capture complex patterns. Finally, we stack several MPNNs for neighborhood interaction, which outputs the final predictions of the offset as follows:

$$\hat{\boldsymbol{x}}_i^{t_1,off} = \psi_{ref}(\{\boldsymbol{z}_i^{t_1}\}_{i\in\mathcal{V}}, \{\boldsymbol{p}_{ij}\}_{(i,j)\in\mathcal{E}}), \tag{11}$$

where $\psi_{ref}$ has a similar architecture to the MPNN-based decoder, but with shallower layers. In the end, the refined predictions can be obtained by combining coarse predictions and offsets:

$$\hat{\boldsymbol{x}}_i^{t_1,ref} = \hat{\boldsymbol{x}}_i^{t_1} + \hat{\boldsymbol{x}}_i^{t_1,off}. \tag{12}$$

The mean square error is minimized for the target observations:

$$\mathcal{L}_{re} = \sum_{i=1}^N ||\hat{\boldsymbol{x}}_i^{t_1,ref} - \boldsymbol{x}_i^{t_1}||_2. \tag{13}$$

In contrast to coarse foresight over multiple timestamps, our refinement module targets a single timestamp, enabling us to further minimize the training loss. Unlike previous one-step predictors (Cao et al., 2023), our FAIR uses future predictions generated by the graph ODE twins for interpolation, which is empirically simpler than extrapolation. Our proposed FAIR employs a two-stage optimization strategy. In the first stage, we train the model to generate coarse long-term trajectories using a multi-task learning framework. In the second stage, we shift our focus to fine-tuning short-term predictions and eliminate additional supervision in Eqn. 9. A comprehensive summary of the learning algorithm can be found in Algorithm 1. This approach can be illustrated as a knowledge distillation framework (Gou et al., 2021; Cho & Hariharan, 2019; Park et al., 2019) where the teacher model (*i.e.*, graph ODE twins) gains broad and generalized knowledge from multiple tasks, while the student model (*i.e.*, refinement module) focuses on the specific target, which is capable of benefiting from the foresight provided by the teacher model as well. In addition, our foresight steps would generate a range of coarse predictions with potential noise, which would serve as the perturbation to the input for the refinement step to release potential overfitting.

## 4.3 PYRAMID-LIKE PROPAGATION

To generate long-term predictions, we alternatively conduct foresight generation and interpolation, resulting in a pyramid-like architecture as illustrated in Figure 1. By employing the graph ODE model, our FAIR is able to capture the long-term dynamics governed by the underlying rules. In addition, during our pyramid-like alternative propagation, our FAIR gains insight into future states which helps in mitigating potential error accumulation. A comprehensive summary of the inference algorithm can be found in Algorithm 2.

Table 1: The RMSE results of the compared methods over different prediction lengths of 1, 50, and all time steps. The best results are displayed in bold. Partial results are consistent with Cao et al. (2023). OOM indicates out-of-memory.

| Dataset | CylinderFlow RMSE ($\times 10^{-3}$) $\downarrow$ | | | Airfoil RMSE ($\times 10^{-1}$) $\downarrow$ | | | DeformingPlate RMSE ($\times 10^{-4}$) $\downarrow$ | | | InflatingFont RMSE ($\times 10^{-4}$) $\downarrow$ | | |
|---|---|---|---|---|---|---|---|---|---|---|---|---|
| | 1 | 50 | all | 1 | 50 | all | 1 | 50 | all | 1 | 50 | all |
| GraphUNets (2019) | 8.09 | 187 | 1650 | 2.93 | 117 | 611 | 2.03 | 5.19 | 54.6 | OOM | OOM | OOM |
| GNS (2020) | 2.61 | 50.7 | 176 | 5.29 | 175 | 639 | 2.23 | 3.21 | 17.2 | 2.14 | 36.9 | 50.7 |
| MeshGraphNet (2021) | 2.26 | 43.9 | 107 | 4.35 | 166 | 695 | 1.98 | 2.88 | 15.1 | 1.95 | 17.8 | 36.5 |
| MS-GNN-Grid (2021) | 2.20 | 27.4 | 84.9 | 2.68 | 122 | 556 | 2.20 | **2.78** | 14.8 | 1.87 | 32.4 | 37.8 |
| Social-ODE (2022) | 2.06 | 36.5 | 99.0 | 2.61 | 135 | 564 | 2.17 | 3.11 | 15.9 | 1.80 | 14.4 | 29.8 |
| BSMS-GNN (2023) | 2.04 | 24.2 | 83.7 | 2.88 | 110 | 421 | 2.87 | 3.18 | 16.9 | 1.77 | 10.8 | 22.0 |
| **FAIR (Ours)** | **1.75** | **22.6** | **62.4** | **1.88** | **95** | **405** | **1.92** | 2.82 | **13.7** | **0.79** | **10.6** | **17.8** |

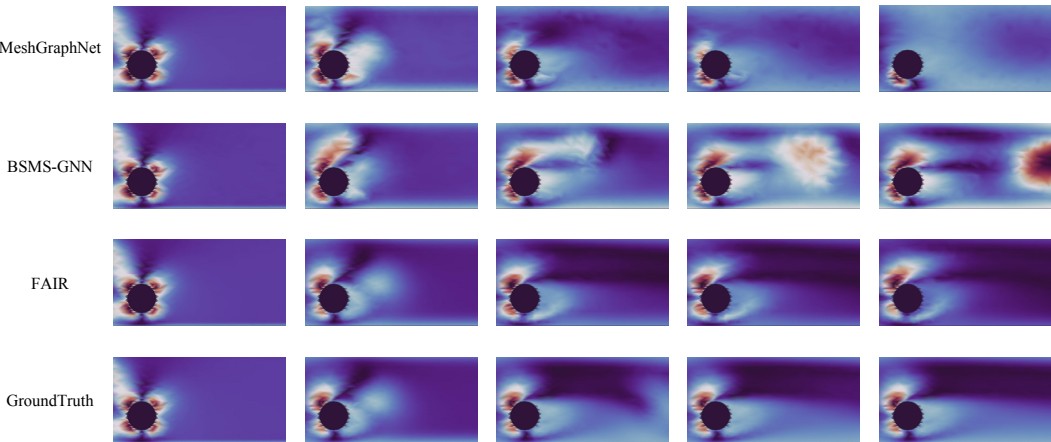

Figure 3: Visualization of different methods on *CylinderFlow* at multiple time steps. We render the velocity in the fluid field with the time steps among $1, 50, 100, 300$ and $500$.

## 5 EXPERIMENTS

### 5.1 EXPERIMENTAL SETUP

**Datasets.** To evaluate our FAIR, we employ four benchmark physics simulation datasets (Pfaff et al., 2021; Cao et al., 2023): 1) *CylinderFlow*, which simulates the flow of an incompressible fluid around a cylinder; 2) *Airfoil*, which focuses on the simulating compressible flow around an airfoil; 3) *DeformingPlate*, which involves the simulation of elastic plate deformation using an actuator; and 4) *InflatingFont*, which depicts the inflation of enclosed elastic surfaces. More detailed information regarding these datasets can be found in Appendix C.

**Baselines.** We compare FAIR with a range of baselines, including neural simulation and a neural ODE method. The neural simulation methods include GraphUNets (Gao & Ji, 2019), GNS (Sanchez-Gonzalez et al., 2020), MeshGraphNet (Pfaff et al., 2021), MS-GNN-GRID (Lino et al., 2021), and BSMS-GNN (Cao et al., 2023). We also adopt a neural ODE approach Social-ODE (Wen et al., 2022). More details of these baselines can be found in Appendix D.

**Implementation.** The root mean square deviation (RMSE) is taken as the metric to evaluate the performance. We vary the prediction lengths to show the performance in both short-term and long-term forecasting tasks. We set the step size $r$ and the future prediction number $L$ as 2 and 3 as default, respectively. More implementation details can be found in Appendix E.

### 5.2 PERFORMANCE COMPARISON

**Quantitative Comparison.** The compared performance is recorded in Table 1. From the results, we can observe that GraphUNets achieve the worse performance compared with the other methods

Table 2: Ablation studies of different variants on four datasets.

| Dataset | CylinderFlow RMSE ($\times10^{-3}$)$\downarrow$ | | | Airfoil RMSE ($\times10^{-1}$)$\downarrow$ | | | DeformingPlate RMSE ($\times10^{-4}$)$\downarrow$ | | | InflatingFont RMSE ($\times10^{-4}$)$\downarrow$ | | |
|---|---|---|---|---|---|---|---|---|---|---|---|---|
| | 1 | 50 | all | 1 | 50 | all | 1 | 50 | all | 1 | 50 | all |
| FAIR *V1* | 1.82 | 24.3 | 75.5 | 1.94 | 102 | 595 | 1.95 | 2.93 | 16.3 | 0.92 | 11.4 | 22.1 |
| FAIR *V2* | 2.03 | 27.2 | 67.5 | 2.31 | 110 | 543 | 2.16 | 3.14 | 16.1 | 1.12 | 12.1 | 21.6 |
| FAIR *V3* | 1.91 | 25.0 | 94.1 | 2.07 | 108 | 602 | 1.99 | 3.16 | 16.9 | 1.21 | 13.6 | 25.0 |
| **FAIR** | **1.75** | **22.6** | **62.4** | **1.88** | **95** | **405** | **1.92** | **2.82** | **13.7** | **0.79** | **10.6** | **17.8** |

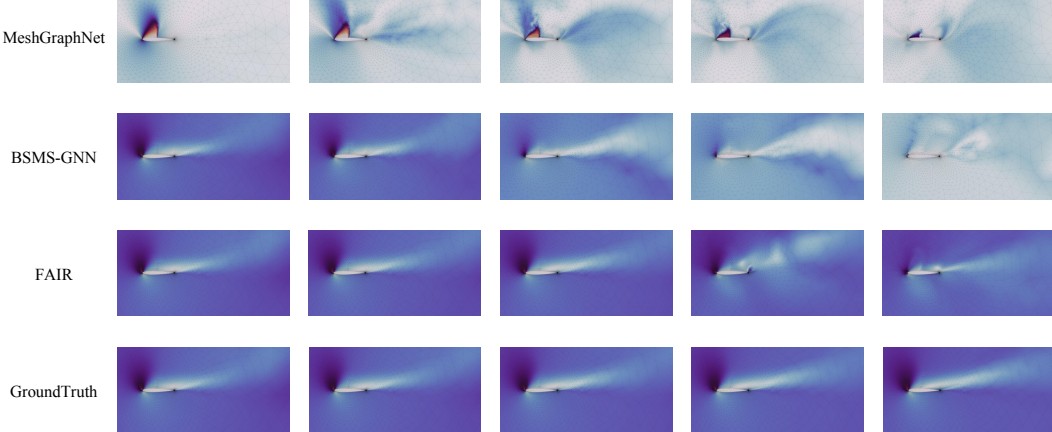

Figure 4: Visualization of different methods on *Airfoil* with the time steps among 1, 50, 100, 150, and 200.

targeting at dynamical system modeling. This indicates the difficulties of the mesh-based physics simulations and we need to design special approaches for this problem. Furthermore, it can be observed that FAIR performs the best across all four datasets in terms of both short-term and long-term forecasting. In particular, compared to the best baseline on each dataset, FAIR achieves an average error reduction of 25.1% in 1-step simulations and 13.9% in all-step simulations. The significant performance improvement of FAIR can be attributed to two factors: (1) The introduction of our graph ODE twins in our foresight step. The graph ODE can significantly reduce the error accumulation in the multi-task learning framework, which also provides future information for short-term predictions; (2) The introduction of our refinement step. It can leverage coarse future predictions to refine short-term predictions with channel aggregation, thus mitigating the underfitting resulting from multi-task learning. Moreover, the performance of our FAIR on MS-GNN-Grid is a little worse than MS-GNN-Grid. The potential reason is the high complexity of DeformingPlate makes it harder to generate accurate foresight, which could deteriorate the model performance.

**Qualitative Comparison.** We also conduct visualization to compare our FAIR with representative baselines and the ground truth. The compared results on *CylinderFlow* and *Airfoil* are shown in Figure 3 and Figure 5.1, respectively. From the results, we can make the following observations: (1) We can find that serious error accumulation occurs for one-step predictors (*i.e.*, MeshGraphNet and BSMS-GNN). For example, in the last frame of Figure 5.1, MeshGraphNet and BSMS-GNN have a huge gap compared with the ground truth. (2) Our model can make precise long-term predictions consistently. The potential reason is that our ODE-based model can capture the continuous dynamics in physical systems and mitigate the error accumulation during propagation. In particular, all the baselines fail to reflect correct flow fields at the last time step while our FAIR can still approximate the ground truth. (3) Our proposed FAIR can also make accurate short-term predictions, which demonstrates that future foresight can benefit short-term predictions as well.

## 5.3 ANALYSIS

**Ablation Studies.** To evaluate the effectiveness of the subcomponents in FAIR, we introduce three model variants as follows: (1) FAIR *V1*, which removes the graph ODE twins module and utilizes

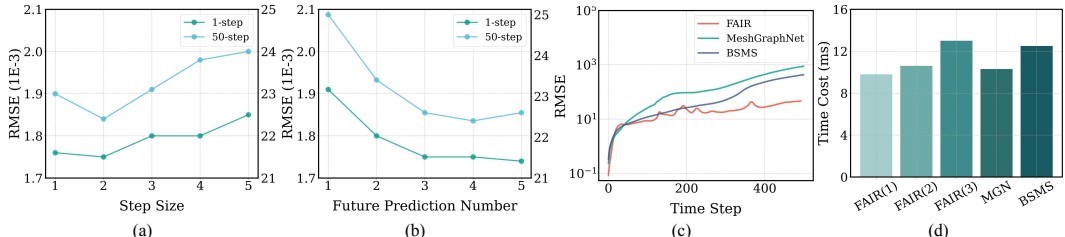

Figure 5: (a), (b) The performance with respect to different step sizes (horizon) $r$ and prediction lengths $L$ on *CylinderFlow*. (c) RMSE of our FAIR and two baselines with respect to different prediction lengths on *Airfoil*. (d) The comparison of the running time of FAIR (with different future prediction numbers), BSMS (BSMS-GNN), and MGN (MeshGraphNet).

one-step predictors for coarse foresight. (2) FAIR *V2*, which removes the refinement from the future module and outputs the results using graph ODE. (3) FAIR *V3*, which removes the multi-task learning framework and involves the next-time predictions in refinement. The compared results are presented in Table 2. From these results, we have the following observations. *Firstly*, by comparing FAIR *V1* with the full model, we can validate that the graph ODE twins module can capture continuous dynamics to provide high-quality future information for accurate predictions. *Secondly*, our full model achieves better performance than FAIR *V2*, which validates that the refinement step is indispensable by preventing potential underfitting in the multi-task learning framework. *Thirdly*, FAIR *V3* performs much worse than the full model, which validates that future information makes a critical contribution to effective mesh-based simulations.

**Sensitivity Analysis.** We investigate the impact of different parameters on the performance of FAIR, *i.e.*, the step size $r$, and the number of future predictions $L$. First, we vary $r$ in $\{1, 2, 3, 4, 5\}$ with the other parameters fixed and the results are shown in Figure 5.3 (a). We can observe that the errors first decrease and then increase as $r$ rises. The potential reason is that when $r$ is small, a larger $r$ can provide a larger horizon while too large $r$ would be far away from our target, which makes the interpolation unreliable. Next, we vary the prediction length $L$ in $\{1, 2, 3, 4, 5\}$, and the results are shown in Figure 5.3 (b). We can observe an error reduction when $L$ rises before saturation, indicating that including more future information boosts model performance.

**Predictions at Different Time Steps.** We compare the prediction errors of our FAIR and two baselines with different time steps in terms of RMSE on *Airfoil*. The results are shown in Figure 5.3 (c). We can find that FAIR exhibits stronger modeling capabilities with relatively lower errors at large time steps while both two baselines suffer from serious error accumulations. This validates that our FAIR utilizes foresight to reduce the error accumulation for long-term predictions.

**Efficiency Analysis.** We analyze the efficiency of MeshGraphNet, BSMS-GNN, and FAIR with varying the number of future predictions $L$. The computational time of one-step predictions on a single NVIDIA A100 GPU is reported. As shown in Figure 5.3 (d), FAIR shows similar efficiency as the baseline methods, demonstrating that FAIR does not introduce major additional time costs. Furthermore, it is observed that the time cost of FAIR increases as $L$ rises. Considering there is a trade-off between efficiency and effectiveness, we set $L$ as 3 in our implementation.

## 6    CONCLUSION

In this paper, we study the problem of long-term mesh-based physics simulations and propose a new approach FAIR to address it. Our FAIR utilizes a future-to-present perspective, which consists of two steps, *i.e.*, foresight and refinement, for accurate simulations. In the first step, we use a graph ODE model that integrates previous states to learn coarse long-term trajectories using a multi-task learning framework. In the second step, we employ a channel aggregation strategy to aggregate the trajectories for refined short-term predictions. Our proposed FAIR can provide accurate long-term trajectories by the alternative propagation of foresight and refinement. We believe that our study provides a brand-new perspective in learning long-term mesh-based simulations. However, our work has a limitation that it cannot make accurate predictions when system states fluctuate in unsteady flows, a complicated scenario encountered in fluid mechanics. In the future work, we plan to expand the proposed FAIR to accommodate more complex simulations in physical and biological applications such as molecular dynamics simulations and protein structural analysis.

ETHICS STATEMENT

We acknowledge that all co-authors of this work have read and committed to adhering to the ICLR Code of Ethics.

REPRODUCIBILITY STATEMENT

To increase reproducibility, we have provided all the details of FAIR in Appendix E. Our code is available at `https://anonymous.4open.science/r/FAIR` anonymously. We will make the code public after the anonymity period. The datasets utilized in this paper are publicly available and representative ones [1][2]. We obey the original settings and divisions without incorporating any additional data. The baseline methods that we utilize are all publicly accessible. The experimental results of the baselines are consistent with Cao et al. (2023).

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

# A    PROOF OF LEMMA 4.1

**Lemma 4.1.** *Suppose we have an FFN $\Phi$ with all absolute values of weights and biases bounded by satisfy $M, B$ respectively. Besides ReLU is adopted as the activation function. Then, our Eqn. 7 has a unique absolutely continuous solution.*

To begin with, we introduce a theorem from (Difonzo et al., 2024).

**Theorem A.1.** *Let $n \in \mathbb{N} \cup \{0\}, \tau \in (0, +\infty), x_0 \in \mathbb{R}^d$ and let $f$ satisfy the following assumptions:*

*(A1) For all $t \in [0, (n+1)\tau]$, $f(t, \cdot, \cdot) \in C\left(\mathbb{R}^d \times \mathbb{R}^d; \mathbb{R}^d\right)$,*

*(A2) For all $(x, z) \in \mathbb{R}^d \times \mathbb{R}^d$, $f(\cdot, x, z)$ is Borel measurable,*

*(A3) There exists $K : [0, (n+1)\tau] \to [0, +\infty)$ such that $K \in L^1([0, (n+1)\tau])$ and for all $(t, x, z) \in [0, (n+1)\tau] \times \mathbb{R}^d \times \mathbb{R}^d$, we have:*

$$\|f(t, x, z)\| \le K(t)(1 + \|x\|)(1 + \|z\|), \tag{14}$$

*(A4) For every compact set $U \subset \mathbb{R}^d$, there exists $L_U : [0, (n+1)\tau] \mapsto [0, +\infty)$ such that $L_U \in L^1([0, (n+1)\tau])$ and for all $t \in [0, (n+1)\tau], x_1, x_2 \in U, z \in \mathbb{R}^d$, we have:*

$$\|f(t, x_1, z) - f(t, x_2, z)\| \le L_U(t)(1 + \|z\|)\|x_1 - x_2\| \tag{15}$$

*Then there exists a unique absolutely continuous solution $x = x(x_0, f)$ to the following system:*

$$\begin{cases} x'(t) = f(t, x(t), x(t-\tau)), & t \in [0, (n+1)\tau] \\ x(t) = x_0, & t \in [-\tau, 0) \end{cases}, \tag{16}$$

*such that for $j = 0, 1, \ldots, n$ we have:*

$$\sup_{0 \le t \le \tau} \|\phi_j(t)\| \le K_j, \tag{17}$$

*where $\phi_{-1}(t) = x_0, \phi_j(t) = x(t + j\tau), \quad for\ j = 0, 1, \ldots, n, K_{-1} := \|x_0\|$ and*

$$K_j = (1 + K_{j-1})\left(1 + \|K\|_{L^1([j\tau, (j+1)\tau])}\right) \cdot \exp\left((1 + K_{j-1})\|K\|_{L^1([j\tau, (j+1)\tau])}\right). \tag{18}$$

*Proof.* Firstly, the function $\Phi$ is learnable FFNs, and the first layer can be represented as:

$$\boldsymbol{m}_1^t = \sigma(\boldsymbol{W}_0 \boldsymbol{y}^t + \boldsymbol{W}_1 \boldsymbol{y}^{t-1} + \boldsymbol{b}_1), \tag{19}$$

where $\boldsymbol{W}_0$ and $\boldsymbol{W}_1$ are two weight matrices, and $\boldsymbol{b}_1$ denotes the biases. The $i$ layers can be written as:

$$\boldsymbol{m}_i^t = \sigma(\boldsymbol{W}_i \boldsymbol{m}_{i-1}^t + \boldsymbol{b}_i), \tag{20}$$

where $\boldsymbol{W}_i$ and $\boldsymbol{b}_i$ are corresponding weights and biases. These functions are continuous and Borel measurable. Therefore, $\Phi$ also satisfies assumptions (A1), (A2).

Secondly, given any inputs $\boldsymbol{y}^t, \boldsymbol{y}^{t-1}$, we can see:

$$\begin{aligned} \|\Phi(\boldsymbol{y}^t, \boldsymbol{y}^{t-1})\| &= \|\boldsymbol{m}_l^t(\cdots \boldsymbol{m}_1^t(\boldsymbol{y}^t, \boldsymbol{y}^{t-1}))\| \\ &\le \|\boldsymbol{m}_l^t(\cdots \boldsymbol{m}_2^t(M\|\boldsymbol{y}^t\| + M\|\boldsymbol{y}^{t-1}\| + B))\| \\ &\le \cdots \le M^l\|\boldsymbol{y}^t\| + M^l\|\boldsymbol{y}^{t-1}\| + \frac{M^l - 1}{M - 1}B \\ &\le L(1 + \|\boldsymbol{y}^t\|)(1 + \|\boldsymbol{y}^{t-1}\|), \end{aligned}$$

where $L = \max\{M^l, \frac{M^l-1}{M-1}B\}$. Thus assumption (A3) is satisfied.

Third, note the ReLU activation function satisfies:

$$|\sigma(x) - \sigma(y)| \le |x - y|. \tag{21}$$

Given any $\boldsymbol{x}_1, \boldsymbol{x}_2, \boldsymbol{z}$, we have:

$$\|\boldsymbol{m}_1^t(\boldsymbol{x}_1, \boldsymbol{z}) - \boldsymbol{m}_1^t(\boldsymbol{x}_2, \boldsymbol{z})\| \le \|\boldsymbol{W}_0(\boldsymbol{x}_1 - \boldsymbol{x}_2)\| \le M\|\boldsymbol{x}_1 - \boldsymbol{x}_2\|. \tag{22}$$

Then, the following inequation holds:

$$\|\boldsymbol{m}_2^t(\boldsymbol{m}_1^t(\boldsymbol{x}_1, \boldsymbol{z})) - \boldsymbol{m}_2^t(\boldsymbol{m}_1^t(\boldsymbol{x}_2, \boldsymbol{z}))\| \le \|W_2(\boldsymbol{m}_1^t(\boldsymbol{x}_1, \boldsymbol{z}) - \boldsymbol{m}_1^t(\boldsymbol{x}_2, \boldsymbol{z}))\| \le M^2\|\boldsymbol{x}_1 - \boldsymbol{x}_2\| \quad (23)$$

Thus, we can conclude:

$$\|\Phi(\boldsymbol{x}_1, \boldsymbol{z}) - \Phi(\boldsymbol{x}_2, \boldsymbol{z})\| \le M^l\|\boldsymbol{x}_1 - \boldsymbol{x}_2\|, \tag{24}$$

which means the assumption (A4) is satisfied.

Then, based on Theorem A.1, we can claim that our graph ODE system Eqn. 7 has a unique absolutely continuous solution. □

## B  ALGORITHM

The training algorithm of our FAIR is summarized in Algorithm 1.

---

**Algorithm 1** Learning Algorithm of FAIR

---

**Input:** The mesh graph $G$, a sequence of observations $G^{t_0:t_0+T} = \{G^{t_0}, \cdots, G^{t_0+T}\}$.
**Output**: Parameters in our FAIR.

1: Initialize model parameters;
2: // *Foresight Step*
3: **while** not convergence **do**
4:     **for** each training sequence **do**
5:         Feed each sample into the graph ODE;
6:         Generate the predictions at the given timestamps using Eqn. 8;
7:         Minimize the mean square error for these timestamps in Eqn. 9;
8:         Update the parameters using gradient descent;
9:     **end for**
10: **end while**
11: // *Refinement Step*
12: **while** not convergence **do**
13:     **for** each training sequence **do**
14:         Generate the predictions at the given timestamps using Eqn. 8;
15:         Generate the refined predictions from Eqn. 12;
16:         Minimize the mean square error for the target in Eqn. 13
17:         Update the parameters using gradient descent;
18:     **end for**
19: **end while**

---

The inference algorithm of our FAIR is summarized in Algorithm 2.

---

**Algorithm 2** Inference Algorithm of FAIR

---

**Input:** The mesh graph $G$, a sequence of observations $G^{t_0:t_0+T} = \{G^{t_0}\}$.
**Output**: Parameters in our FAIR.

1: $t = t_0$
2: $\hat{\boldsymbol{X}}^{t_0, ref} = \boldsymbol{X}^{t_0}$
3: **while** $t < t_0 + T$ **do**
4:     // *Foresight Step*
5:     Feed $\hat{\boldsymbol{X}}^{t, ref}$ into the graph ODE;
6:     Generate the predictions $\hat{\boldsymbol{X}}^{t+1}, \hat{\boldsymbol{X}}^{t+1+r}, \cdots, \hat{\boldsymbol{X}}^{t+1+rL-r}$ using Eqn. 8;
7:     // *Refinement Step*
8:     Generate the refined predictions $\hat{\boldsymbol{X}}^{t+1, ref}$ from Eqn. 12;
9:     $t \leftarrow t + 1$
10: **end while**

---

## C   DATASET DETAILS

Four physics simulation benchmark datasets are utilized to evaluate our proposed FAIR and the compared baselines with details listed in Table 3.

*CylinderFlow* simulates the incompressible Navier-Stokes flow of water around a cylinder on a fixed 2D Eulerian mesh generated by COMSOL (Multiphysics, 1998). This mesh has an irregular structure with varying edge lengths in different regions. The simulation consists of $600$ time steps, with an interval of $0.01$s between each step. Node attributes in the system include mesh position, node type, velocity, and pressure. Node types can be divided into three different categories in fluid domains, *i.e.*, fluid nodes, wall nodes, and inflow/outflow boundary nodes. We predict the velocity values in both directions.

*Airfoil* simulates the aerodynamics around the cross-section of an airfoil wing for compressible Navier-Stokes flow by SU2 (Economon et al., 2016). As the edge lengths of the mesh range between $2 \times 10^{-4}$m to 3.5m, the mesh structure is highly irregular. Each trajectory containing $5,200$ nodes has $500$ time steps with an interval of $0.008$s. Node attributes include mesh position, node type, velocity, pressure, and density. We aim to predict the velocity, density, and pressure in the future.

*DeformingPlate* is a hyper-elastic plate in the structural mechanical system, deformed by a kinematic actuator, simulated with a quasi-static simulator COMSOL. Each trajectory has $400$ time steps with $1,200$ nodes average. A one-hot vector for each type of node distinguishes actuators from plates in the Lagrangian tetrahedral mesh. In addition, node type, position, and velocity are fed to predict the whole trajectories.

*InflatingFont* is from BSMS-GNN (Cao et al., 2023), including $1,400$ $2 \times 2$-character matrices in Chinese. *InflatingFont* has more complex geometric shapes, 2 to 8 times the number of nodes, and 70 times the number of contact edges. We aim to predict the future position of every mesh node.

For each dataset, the train/val/test splits follow the recent work (Cao et al., 2023).

Table 3:   All of our four datasets are listed in detail. The system describes the underlying PDE: hypere-lastic flow, or a compressible or incompressible Navier-Stokes flow. Simulation data is generated using a solver. In *DeformingPlate* and *InflatingFont*, there is no time step since it is a quasi-static simulation.

| Dataset | Nodes (avg) | Edge (avg) | Type | Steps |
|---|---|---|---|---|
| CylinderFlow | 1885 | 5424 | Eulerian | 600 |
| Airfoil | 5233 | 15449 | Eulerian | 500 |
| DeformingPlate | 1271 | 4611 | Lagrangian | 400 |
| InflatingFont | 13177 | 39481 | Lagrangian | 100 |

## D   BASELINE DETAILS

We compare our FAIR with a range of state-of-the-art methods, *i.e.*, GraphUNets (Gao & Ji, 2019), GNS (Sanchez-Gonzalez et al., 2020), MeshGraphNet (Pfaff et al., 2021), MS-GNN-GRID (Lino et al., 2021), Social-ODE (Wen et al., 2022) and BSMS-GNN (Cao et al., 2023). Their details are elaborated as follows:

- GraphUNets (Gao & Ji, 2019) proposes new pooling and unpooling operations, which can be implemented in an UNet-style architecture (Ronneberger et al., 2015). We have replaced the original GCN layers into our message passing layers following (Cao et al., 2023).

- GNS (Sanchez-Gonzalez et al., 2020) is the pioneering work on physical simulations, which leverage graphs to depict systems and model dynamics using message passing neural networks. This work demonstrates that graph neural networks have the ability to capture long-range interactions. We employ 15 message passing layers as in Cao et al. (2023).

- MeshGraphNet (Pfaff et al., 2021) is an effective framework for mesh-based physical simulations, which combine graph neural networks and re-mesh techniques to learn the dynamics for next-time

predictions. Based on the basic node modeling of graph networks, MeshGraphNet introduces additional edge encoders. The edge representation is updated during each MeshGraphNet layer. We also employ 15 message passing layers as in Cao et al. (2023).

- MS-GNN-GRID (Lino et al., 2021) is a representative work for those building the hierarchy with spatial proximity, which introduces a novel multi-scale graph neural network model, designed to enhance and accelerate predictions in continuum mechanics simulations. Following (Lino et al., 2021; Cao et al., 2023), MS-GNN-GRID is implemented using the finest edge encoder, an additional aggregation module for node and edge representations, and a node returning module.

- Social-ODE (Wen et al., 2022) is a latent ordinary differential equation generative model, which can understand and predict dynamics from irregularly-sampled partial observations with underlying graph structures. In our implementation, we use a similar structure paradigm with our approach, *i.e.*, Encoder-ODE-Decoder pattern, where both the encoder and decoder consist of 7 layer networks for node and edge modeling.

- BSMS-GNN (Cao et al., 2023) is a framework that introduces a bi-stride pooling strategy for large-scale physical simulations, addressing existing challenges associated with scaling complexity, over-smoothing, and incorrect edge introductions. BSMS-GNN follows the design paradigm of UNet (Ronneberger et al., 2015). We adhere to the original network configurations and utilize several UNet layers for experiments.

## E    IMPLEMENTATION DETAILS

In this paper, we present an extensive series of experiments leveraging the frameworks of PyTorch (Paszke et al., 2017), PyG (Fey & Lenssen, 2019), and TorchDiffEq (Kidger et al., 2021). To ensure fairness, we implement our approach with a publicly available codebase by BSMS-GNN (Cao et al., 2023). We execute all experiments on a single A100 GPU, including the speed test. To maximize training efficiency, we set the batch size across all experiments at the maximum level supported by the GPU memory. Our optimization strategy includes the use of Adam optimizer, with a learning rate set at $1e - 4$, with an exponential learning rate decay strategy.

Following BSMS-GNN, we apply Gaussian noise to each original trajectory at the start of every epoch, aiming to enhance the adaptability of the model to process noisy inputs. Furthermore, to maintain fairness, we set the layer number for both encoder and decoder to 7, while MeshGraphNet and BSMS-GNN both have 15 layers. Each layer includes the nodal encoder, processor, and nodal decoder, all activated by ReLU and embedded within two hidden-layer MLPs. The MLPs have a residual connection, while a LayerNorm normalizes all MLP outputs except for the nodal decoder. Our code is available at `https://anonymous.4open.science/r/FAIR` anonymously. We will make the code public after the anonymity period.

## F    MORE RESULTS

### F.1    PREDICTIONS AT DIFFERENT TIME STEPS

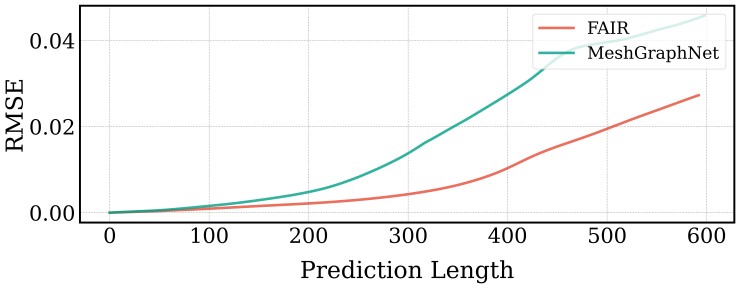

Figure 6: RMSE of our FAIR and MeshGraphNet with respect to different prediction lengths on *CylinderFlow*.

Figure 6 records the prediction errors of our proposed FAIR and MeshGraphNet with different time steps in terms of RMSE on *CylinderFlow*. From the results, we can observe that our proposed

FAIR demonstrates stronger modeling capabilities with relatively lower errors at large-time steps. In contrast, MeshGraphNet suffers from serious error accumulations, with the prediction error about twice as large as ours at the final time step.

## F.2 VISUALIZATION

Figure 7 shows more visualization at a range of time steps on *CylinderFlow*. Here we utilize a different instance to show the results at the time steps among $\{1, 10, 50, 100, 300, 400\}$. From the results, we can validate the superiority of the proposed FAIR again.

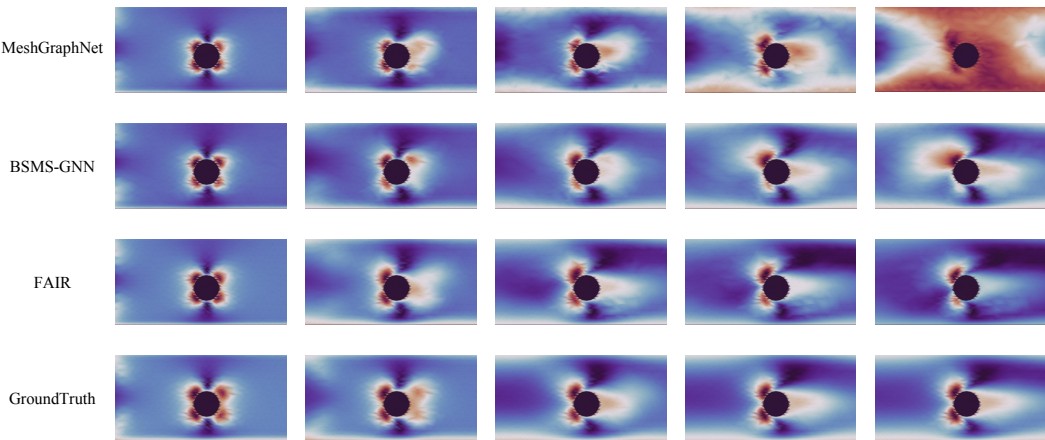

Figure 7: Visualization of *CylinderFlow* at multiple time steps.

## F.3 INFLUENCE OF DIFFERENT STEP SIZES

Figure 8 shows the results with respect to different step sizes $r$ on *Airfoil*. From the results, it can be found that the prediction errors first decrease and then increase, which achieves the minimum when $r$ is around 2. This observation is consistent with that in Sec. 5.3.

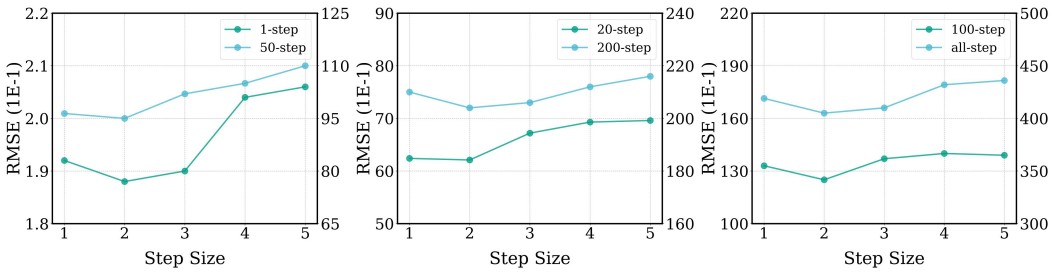

Figure 8: The performance with respect to different step sizes $r$ at different time steps on *Airfoil*.

## F.4 INFLUENCE OF FUTURE PREDICTION NUMBERS

Figure 9 records the results with respect to different future prediction numbers $L$ on *Airfoil*. From the results, we can observe that prediction errors would decrease generally till saturation and the errors would not dramatically decrease when $L$ is over 3 in most cases. Due to the trade-off between effectiveness and efficiency, we set $L$ to 3 as default.

## F.5 VISUALIZATION OF ERRORS

Figure 11 visualizes the prediction errors of different approaches compared with the ground truth on *CylinderFlow*, respectively. From top to bottom, we show the results at the time steps 1, 10, 50,

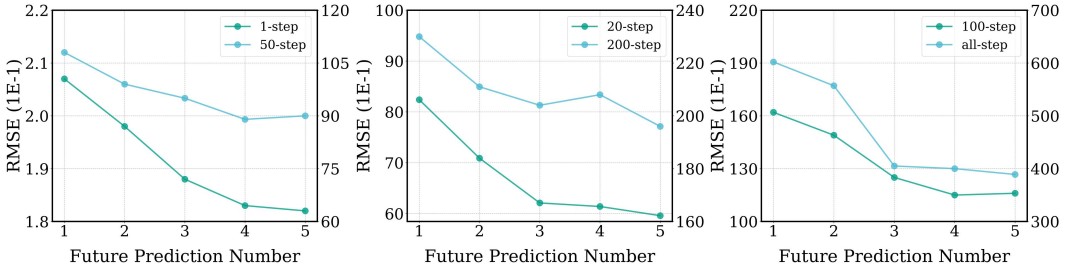

Figure 9: The performance with respect to different prediction lengths $L$ at different time steps on *Airfoil*.

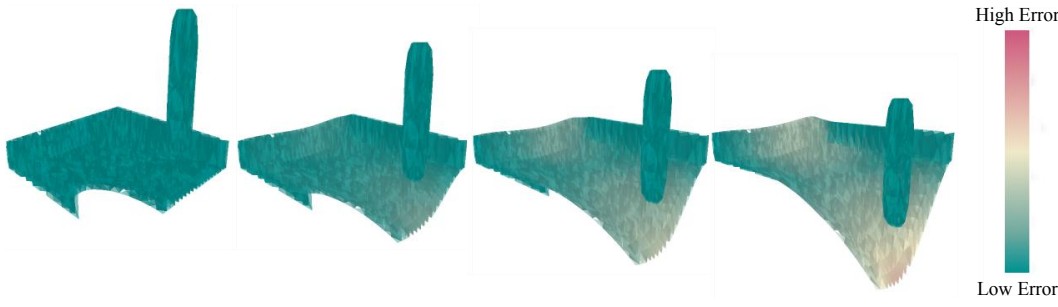

Figure 10: Visualization of FAIR on 3D dataset *DeformingPlate*, with the time steps among $1, 100, 200$ and $300$.

$100, 300$ and $400$. We can observe that although both baselines and our FAIR can make accurate short-term predictions with limited prediction errors, these baselines would suffer from serious error accumulation in long-term forecasting. Figure 12 visualizes the prediction errors on *Airfoil* and we can find that our FAIR is consistently superior to the compared baselines. Moreover, Figure F.4 shows the difference on 3D dataset *DeformationPlate*.

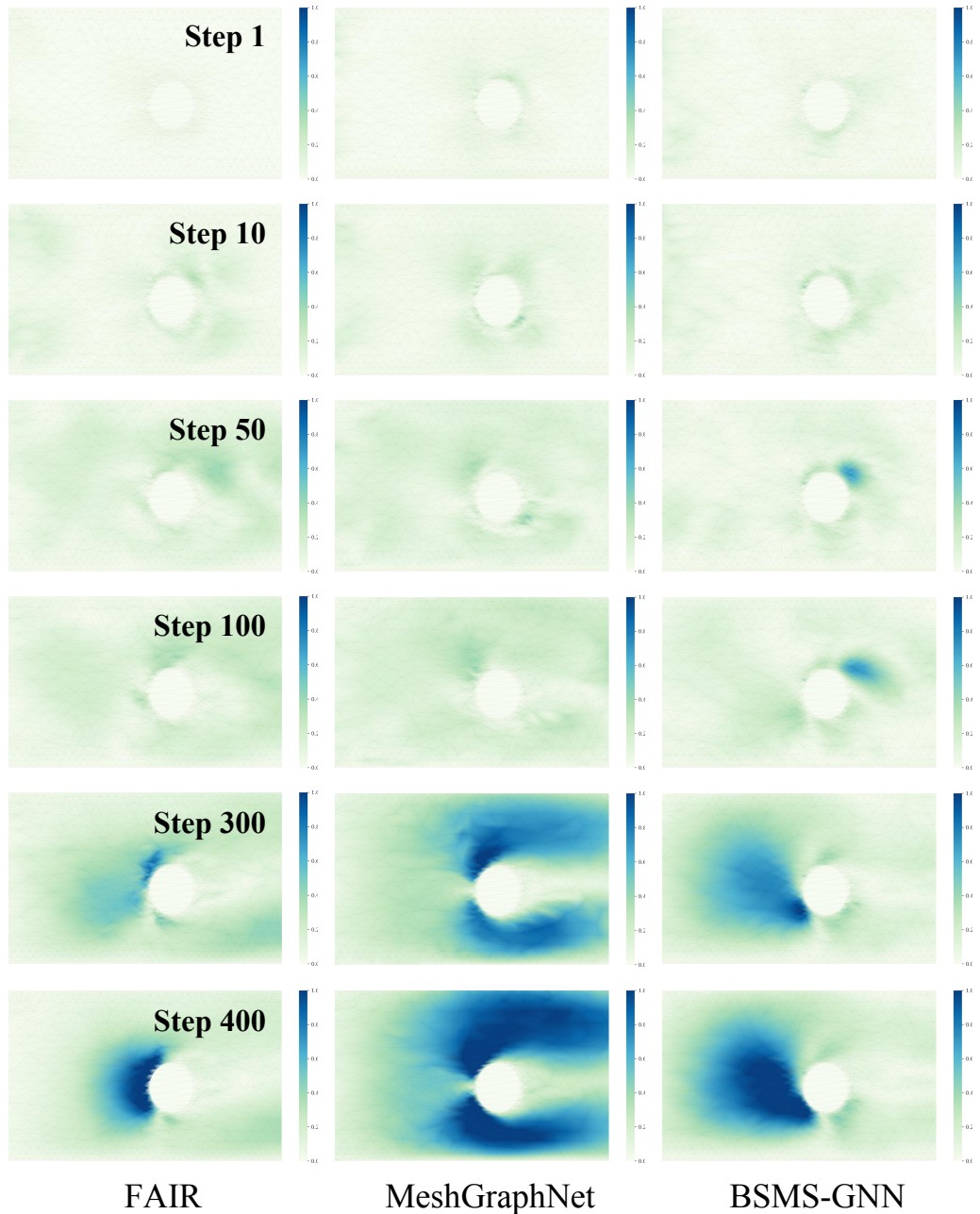

Figure 11: Visualization of error on *CylinderFlow* with multiple time steps.

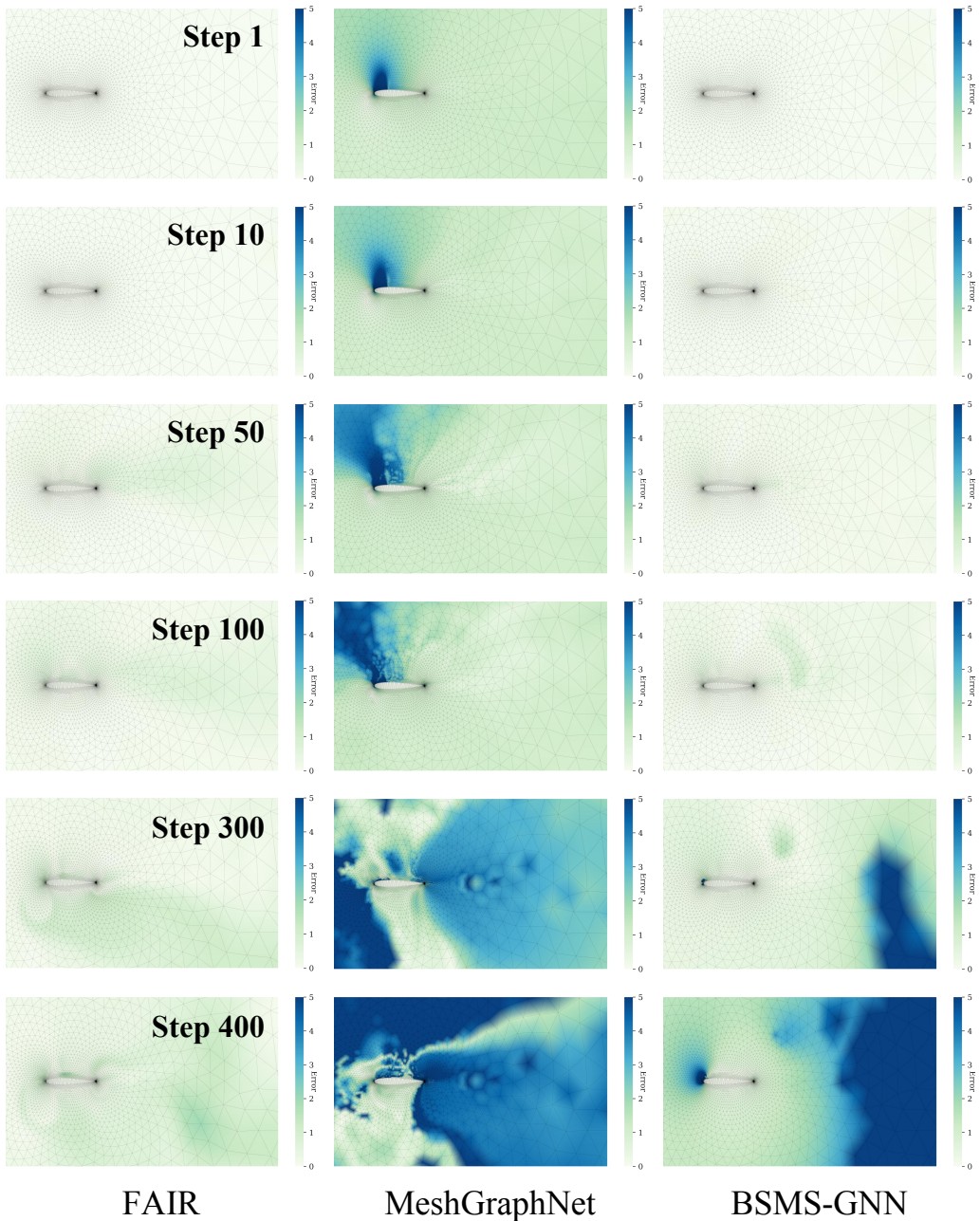

Figure 12: Visualization of error on *Airfoil* at multiple time steps.

