# OpenReview forum: "Learning from the Future: Improve Long-term Mesh-based Simulation with Foresight"
_ICLR.cc/2024/Conference — Submitted to ICLR 2024_

### Official Review · Reviewer_wp4W · 2023-10-26

**Soundness:** 2 fair
**Presentation:** 1 poor
**Contribution:** 2 fair
**Rating:** 3
**Confidence:** 4

**Summary:**

This paper propose a coarse-refine framework for long-term simulation. The model first predicts future states, providing auxiliary information for further refinement of the next step predictions. The results show lower errors comparing with baseline GNNs. The errors of long-term predictions accumulate less faster than baselines.

**Strengths:**

* The proposed method obtains lower errors for both short-term and long-term predictions.

**Weaknesses:**

The writing of the paper is poor. Several examples are as follows:
1. Many duplicated expressions. E.g. Section 4.1 "which are flexible to produce flexible outputs at any given timestamp".
2. Lack proper explanations and details. E.g., any explanation for the "multi-task learning" that is used to train the model?
3. Poor organizations and missing labels for figures. E.g., in Fig 5, what do the "a,b,c,d" refer to? The color in Fig 5 is not clear to distinguish different settings.

**Questions:**

Can the coarse-refine framework apply to the baselines? Will the proposed framework benefit them also?

---

> ### Author Response · Authors · 2023-11-22
> **Response to Reviewer wp4W**
>
> We are truly grateful for the time you have taken to review our paper and your insightful review. Here we address your comments in the following.
>
> > Q1. Many duplicated expressions. E.g. Section 4.1 "which are flexible to produce flexible outputs at any given timestamp".
>
> A1. Thanks for your comment. We have corrected the typos and corrected the expressions.
>
> > Q2. Lack of proper explanations and details. E.g., any explanation for the "multi-task learning" that is used to train the model?
>
> A2. Thanks for your comment. Here, we generate predictions with different prediction lengths, each of which corresponds to a task. These tasks share the same encoder and decoder. We have included the explanation to clarify the introduced concept.
>
>
> > Q3. Poor organizations and missing labels for figures. E.g., in Fig 5, what do the "a,b,c,d" refer to? The color in Fig 5 is not clear to distinguish different settings.
>
> A3. Thanks for your comment. We have included "a,b,c,d" in Figure 5 and make the figure more clear.
>
> > Q4. Can the coarse-refine framework apply to the baselines? Will the proposed framework benefit them also?
>
> A4. Thanks for your comment. It would be out-of-memory for any baselines since they adopt the rollout methods for long-term predictions, which could include a large computation cost to output a range of predictions at the same time. In contrast, our FAIR utilizes a graph ODE to generate **coarse predictions** to release the efficiency issue, which makes our framework feasible.
>
>
> In light of these responses, we hope we have addressed your concerns, and hope you will consider raising your score. If there are any additional notable points of concern that we have not yet addressed, please do not hesitate to share them, and we will promptly attend to those points.

---

> ### Author Response · Authors · 2023-11-23
> **Thank you for your invaluable feedback!**
>
> Dear Reviewer,
>
>
> Thank you for your invaluable feedback. As the deadline for the author-reviewer discussion phase is approaching, we hope to make sure that our response sufficiently addressed your concerns regarding the writing, as well as the revised version of our paper. We hope this could align with your expectations and positively influence the score. Please do not hesitate to let us know if you need any clarification or have additional suggestions.
>
> Best Regards,
>
> Authors

---

### Official Review · Reviewer_L1zc · 2023-10-30

**Soundness:** 3 good
**Presentation:** 3 good
**Contribution:** 3 good
**Rating:** 6
**Confidence:** 4

**Summary:**

The paper presents a new meshed-based physics simulation approach named FAIR. The key idea is to use a coarse-to-fine learning paradigm to reduce the accumulated errors in long-term prediction. The model uses a continuous graph ODE module for generating coarse long-term predictions, subsequently refining them using interpolation techniques to derive short-term predictions.

FAIR is extensively evaluated on four benchmarks, showing superior performance in both short-term and long-term forecasting when compared to various baselines. The paper also provides ablation studies, sensitivity analysis, and visualization to validate the effectiveness of the proposed model.

In general, the paper presents a well-motivated approach that delivers impressive empirical results.

**Strengths:**

1. Clarity: This paper is clearly written and easy to follow.
2. Novelty: Unlike previous one-step prediction methods, the paper introduces a two-stage, foresight-and-interpolation method to improve the long-term prediction results of learning-based physics simulators. Overall, the architectural design of graph ODE and channel aggregation is reasonable.
3. Experiments: The proposed model showcases impressive results over existing models.

**Weaknesses:**

1. The proposed model is closely related to GMR-GMUS (Han et al., 2022), which similarly aims to enhance long-term prediction in mesh-based simulators. Notably, GMR-GMUS achieves long-term predictions for 400 time steps in the case of Cylinder Flow. Including a comparison between FAIR and this approach in the experiments would be valuable.
2. In Table 1, for the DeformingPlate experiment with prediction lengths of 50, MS-GNN-Grid has the best result with an RMSE of 2.78. It would be beneficial to highlight this result in bold and provide some discussions hopefully.

Minor issues:

3. In Figure 5, it would be helpful to annotate the four sub-figures with (a-d) to improve reference.
4. In the last paragraph of Section 4.2: 'the inference algorithm can be found in Algorithm 1' should be revised to 'the learning algorithm...'.
5. The paper includes duplicate notations for $L$, referring to both the prediction length and the number of stacked layers.

References:
Han, et al. Predicting physics in mesh-reduced space with temporal attention. ICLR 2022.

**Questions:**

1. I am a little confused with the equation $t_l = t_0 + rl − r + 1$ in Section 4.1. What does step size mean in the paper?
2. In Section 4.2, it would be beneficial if the authors could provide an explanation for their choice to compute the offset in the observation space, as opposed to calculating an updated latent vector in the latent space.
3. In Eq.(11), does the notation $\{z_i^{t_1}\}_{i \in \mathcal{V}}$ indicate that the offset of node $i$ is computed using representations from all nodes in the set $\mathcal{V}$?

---

> ### Author Response · Authors · 2023-11-22
> **Response to Reviewer L1zc**
>
> We are truly grateful for the time you have taken to review our paper, your insightful comments and support. Your positive feedback is incredibly encouraging for us! In the following response, we would like to address your major concern and provide additional clarification.
>
> > Q1. The proposed model is closely related to GMR-GMUS (Han et al., 2022), which similarly aims to enhance long-term prediction in mesh-based simulators. Notably, GMR-GMUS achieves long-term predictions for 400 time steps in the case of Cylinder Flow. Including a comparison between FAIR and this approach in the experiments would be valuable.
>
>
> A1. Thanks for your comment. We are willing to compare the method with our FAIR but their source code is not publicly available. Due to the limited time, we are unable to finish re-implementing their code according to the algorithms. We therefore only cited and discussed these important baselines in Section 2 in our updated paper.
>
>
> > Q2. In Table 1, for the DeformingPlate experiment with prediction lengths of 50, MS-GNN-Grid has the best result with an RMSE of 2.78. It would be beneficial to highlight this result in bold and provide some discussions hopefully.
>
> A2. Thanks for your comment. We have highlighted the results and included the following discussion:
> "Moreover, the performance of our FAIR on MS-GNN-Grid is a little worse than MS-GNN-Grid. The potential reason is the high complexity of DeformingPlate makes it harder to generate accurate foresight, which could deteriorate the model performance."
>
> > Q3. In Figure 5, it would be helpful to annotate the four sub-figures with (a-d) to improve reference.
>
> A3. Thanks for your comment. We have included these annotations as you suggested.
>
> > Q4. In the last paragraph of Section 4.2: 'the inference algorithm can be found in Algorithm 1' should be revised to 'the learning algorithm...'.
>
> A4. Thanks for your comment. We have revised the sentence as you suggested.
>
>
> > Q5. The paper includes duplicate notations for L, referring to both the prediction length and the number of stacked layers.
>
> A5. Thanks for the comment. We have corrected this by including letters $L^p$ and $L^s$.
>
> > Q6. In Section 4.1. What does step size mean in the paper?
>
> A6. Thanks for the comment. The step size means the interval for coarse predictions, which is $r$ in $t_{l}=t_{0}+r l-r+1$.
>
> > Q7. In Section 4.2, it would be beneficial if the authors could provide an explanation for their choice to compute the offset in the observation space, as opposed to calculating an updated latent vector in the latent space.
>
> A7. Thanks for the comment. Firstly, utilizing observation space is more efficient due to low dimension. Moreover, the calculation in the observation space is more close to interpolation.
>
>
>
>
> > Q8. In Eq.(11), does the notation indicate that the offset of a node is computed using representations from all nodes in the set?
>
>
> A8. Thanks for the comment. For efficiency consideration, we only introduce the node itself without neighborhood information, which has been utilized for coarse prediction generation.
>
> Thanks again for appreciating our work and for your constructive suggestions. Please let us know if you have further questions.

---

### Official Review · Reviewer_vkK1 · 2023-10-31

**Soundness:** 1 poor
**Presentation:** 1 poor
**Contribution:** 2 fair
**Rating:** 1
**Confidence:** 4

**Summary:**

This paper studies a two-stage approach to predicting physical systems with graph networks. The method aims to provide long-term stability on complex non-linear dynamics. The first stage conducts temporally coarse predictions of multiple system states. To this end, a neural ODE approach was embedded in the graph structure. In principle, this allows for flexible timesteps by the means of the ODE integrator. The second stage then takes the series of frames to refine the first prediction. An autoregressive rollout of both stages then yields the long-term dynamics of the system. The method is evaluated against a series of baselines on steady-state physics scenarios.

**Strengths:**

### Originality
The paper introduces a method that combines a prediction model with a subsequently applied refiner. There is some novelty in applying such an approach to physics predictions with graph networks. However, the fact that similar procedures have been studied in video prediction (see e.g. *“Flexible Diffusion Modeling of Long Videos”* by Harvey et al., NeurIPS 2022) should be denoted in the paper. Furthermore, similar methods exist for PDE predictions (*“DYffusion: A Dynamics-informed Diffusion Model for Spatiotemporal Forecasting”* by Cachay et al., NeurIPS 2023, *“PDE-Refiner: Achieving Accurate Long Rollouts with Neural PDE Solvers”* by Lippe et al., NeurIPS 2023). However, the latter methods are recent and can be considered contemporary, i.e. a direct comparison against them is not explicitly required.

### Clarity
The majority of the paper apart from section 4 is clearly phrased. The line of argumentation, while not coherent in the bigger picture, can be understood decently. The visual quality of some figures is quite good. Especially Fig. 2 helps in understanding the model architecture.

### Quality and Soundness
The authors compare their method to many baselines and the choice of these baselines seems sound with only few exceptions (Social-ODE stands out as a rather odd choice). However, the results obtained with the baselines raise serious concerns about the implementation of these methods as well as the general experimental setup, especially for the MeshGraphNet. Source code is provided for the paper, however no baseline methods are included, at least at a quick glance (I did not investigate the source code in much detail).

### Significance
The authors target improvements in the long-term stability of physics predictions. This is indeed an open problem and of high significance for autoregressive architectures. The method introduced in the paper touches upon a series of changes that could be promising avenues toward achieving this stability. For instance, the shown experiments study variations in the length of the trajectory used in training the predictor ($L$), as well as the relative size of the prediction timestep ($r$). An ablation study in the paper also confirms that the addition of the refiner network improves the accuracy of the method across the entire studied time horizon.

**Weaknesses:**

### Presentation

**P1:**
The line of argumentation is in some cases weird, unclear, vague, or lacks citations, especially in section 3 and 4:
- “Given these limitations….” (difficult to understand, top of page 2)
- “Our graph ODE model is flexible to generate…” (sound weird, middle of page 2)
- “The model would be evaluated…” (sound weird, end of section 3.1)
- “However, they are inferior in modeling…” (unfounded/missing citation, end of section 3.2)
- “To capture the hierarchy structure among meshes…” (unclear, end of section 3.3)
- “As a foundational step, it is…” (sounds weird, begin of section 4.1)
- “To learn interacting dynamics…” (very unclear, begin of section 4.1)
- “Furthermore, we notice that data-driven…” (unfounded/missing citation, middle of section 4.1)
- “Firstly, our FAIR…, Secondly, our approach…” (unclear, end of section 4.1, very similar at the begin of section 4.3)
- “multi-task learning framework” (unclear, what other task?, frequently across the paper)

**P2:**
Section 4 describing the methodology has structural issues, is generally difficult to follow, and it is not clear to me how the proposed method actually works in detail. Similar problems can be found across the other parts of section 4, but here an example issue from section 4.2: The term interpolation might be misleading for the refinement step. To my understanding, the refinement is mainly used to improve the outputs at discrete timesteps where a coarse prediction already exists. This refinement seems to be necessary to stabilize recurrent applications. Please clarify the refinement procedure in the paper.

**P3:**
Presentation problems for the different evaluations are listed in combination with the corresponding evaluation problem **E** below.

**P4:**
Minor issues and Typos:
- Fig. 1 is broken across a range of PDF viewers on Ubuntu (Firefox, Envince) and also does not print properly. It was only possible to display in Google Chrome
- “unstructured surfaces”; I think you mean unstructured grids
- 2x which learns coarse... (directly before section 3)


### Evaluations

**E1 - Datasets:**
Please provide some physical characteristic numbers, i.e. Reynolds and Mach numbers, for the Navier-Stokes cases. This would greatly improve the comparability of this work to future research. In particular, it would be interesting to know whether the Aerofoil case includes the transonic regime. The conclusion section briefly mentions that the method currently fails when predicting non-steady dynamics. Three out of the four datasets were originally introduced in *Pfaff et al. (2021)*, and include a variety of steady-state as well as transient cases. To me, it is unclear whether this full dataset was used in training and testing, or whether only a small subset with steady-case simulations was used.

**E2 - Table 1:**
- I have some general concerns regarding the evaluation fairness here. Why would a long term prediction+interpolation approach work better on single step predictions across data sets compared to all other methods? To me that is highly unintuitive and not very plausible.
- Only after closer inspection of the appendix and the reproducibility statement (*“The experimental results of the baselines are consistent with Cao et al. (2023).”*) it becomes apparent that the entirety of the baseline results in Tab. 1 are directly copied from *Cao et al. (2023)*. Nevertheless, it is stated throughout the paper and appendix that the authors use their implementation of these methods. In my opinion, this is highly problematic and I want to urge the authors to clearly state the source of the values in the table heading. A fair evaluation against FAIR would require the usage of one consistent setup for all methods, as it is possible that e.g. details in the error aggregation change results.
- Furthermore, I have serious doubts regarding the soundness of these quantitative evaluations. In three out of four cases, the paper reuses the datasets from *Pfaff et al. (2021)* (i.e. CylinderFlow, Airfiol, DeformingPlate), and proceeds to use the MeshGraphNet trained within that publication as the main comparative baseline for visualizations. However, all results in Tab. 1 significantly differ (sometimes by more than an order of magnitude) from the ones reported by *Pfaff et al. (2021)* for the very same datasets (e.g. CylinderFlow RMSE on 50 steps: 43.9e-3 here vs. 6.3e-3 from *Pfaff et al.*). At the same time, the limited information in the appendix suggests that exactly the same architecture (e.g. number of message passing layers) was used as in *Pfaff et al. (2021)*. No explanation is given for these large differences in the error metrics.

**E3 - Fig 3 and Fig. 4:**
- It is unclear which quantity is shown in the figures (velocity amplitude? x or y component?). In general, I would recommend visualizing the vorticity for these flows. Additionally, the flow fields in Fig 3. seem to have strong artifacts, even in the smooth ground truth. There appears to be an issue with the visualization procedure.
- The scale of the shown quantity is not mentioned, please add a colorbar and use the same scale for all plots. Fig 4. even lacks time step descriptions. For both cases, adding a Reynolds number would be helpful to assess the complexity.
- The paper aims to *“enhance the capacity to capture non-linear complex patterns”* and *“capture the long-term dynamics”*, yet the figures visualize a simulation converging to a steady state (i.e. a simulation state with zero temporal derivatives). Visualizing these particular cases fails to emphasize the supposed strength of the presented method, which targets complex non-linear dynamics. In the ground truth, the flows seems to reach a steady state after t=100 for Fig. 3 (and is directly steady from the leftmost frame in Fig. 4), so obviously long-term predictions with interpolation work better than any autoregressive rollout, since no interpolation has to be performed.
- Considering these issues, it would be highly necessary to show a diverse set of videos of predictions, as well as unsteady example cases for a convincing argumentation. In videos, I would expect temporal coherence issues in the predictions for unsteady cases as well, caused by the prediction + interpolation setup.
- In the original MeshGraphNet visualizations (https://sites.google.com/view/meshgraphnets#h.p95069gdfedm) all the prediction results are a lot closer to the ground truth than what is shown here. This raises concerns about the reimplementation and training of the model (as both architecture and data set are claimed to be identical with the work from *Pfaff et al. (2021)*).
- In Fig. 4: The MeshGraphNet visualization seems to be based on an entirely different case compared to all others. Judging from the results shown in *Pfaff et al. (2021)*, it seems unlikely that this architecture fails even on a 1-step prediction, while simultaneously reproducing a trajectory from a different part of the dataset.

**E4 - Fig. 5:**
- I assume this study was once again performed with data that only converges to a steady state solution? This raises the same issues discussed above in *E3*. Prediction videos corresponding to the shown results (especially for c)) would be required once again.
- a): For which fixed L was the sweep across r computed?
- a) and b): Apart from the model architecture, there are differences between FAIR and the baselines in terms of training procedures. I think it is important to emphasize the relative improvement of variations in $r$ and $L$ over the “standard” approach of choosing $r=1$ and $L=1$. While setting $r=2$ lowers the RMSE by ~2%, choosing $L=4$ is far more important and reduces errors by ~12% (evaluated for 50 steps). Crucially, $r$ and $L$ can also be realized in the baselines. The $r$ could similarly be a larger timestep in other methods, while $L$ is equivalent to an unrolled trajectory. In fact, matching an entire training trajectory instead of just the next step is known to yield better results, especially over long horizons (see e.g. *“DPM: A deep learning PDE augmentation method with application to large-eddy simulation”* by Sirignano et al., JCP, 2020;  *“Machine learning–accelerated computational fluid dynamics”* by Kochkov et al., PNAS, 2021;  *“Solver-in-the-loop: Learning from differentiable physics to interact with iterative PDE-Solvers”* by Um et al., NeurIPS, 2020). The long-term stability of FAIR might be entirely due to choosing a large $L$. I would suggest testing the $L=1$ model on unsteady, longer inference horizons to see whether this holds true in experiments. Additionally, training the baselines with the same $r$ and $L$ as chosen for FAIR would yield better comparability.
- c): I would suggest plotting this in log-scale.

**E5:**
To summarize, the scope of this paper is limited to steady-state predictions, where no temporal changes should occur once the solution converges. In this context, it is questionable whether fine temporal resolutions are at all desirable. Thus, the temporal interpolation capabilities have little value in the current state of the paper. Some existing approaches can even directly predict the steady state solution without intermediate steps at all (e.g. *“Deep Learning Methods for Reynolds-Averaged Navier-Stokes Simulations of Airfoil Flows”*, Thuerey et al., AIAA Journal, 2020). Due to the steady-state test cases, the evaluations in this paper mainly focus on two qualities:
1) whether a particular model architecture can calculate the steady state in the first place, and
2) whether the model can stop the solution from drifting over long horizons when the steady state is already reached.

These two aspects should be clearly emphasized across the paper. To achieve a fair comparison, I would suggest training the baseline models on the same timestep as FAIR (i.e. with r=2) to reach a comparable level of recurrency between the models. In addition, investigating unsteady cases for which most baselines were designed seems necessary as well. Alternatively, comparing to methods that directly predict the steady state solution would be an option, however I would expect a recurrent approach to have drawbacks in this scenario.

### Summary
Overall, in addition to substantial presentation problems, this paper has fundamental issues across all performed evaluations and results, that could only be fixed with a major revision well beyond the scope of a conference rebuttal. Thus, I would oppose the publication of this paper in its current state and recommend an overall score of strong reject.

**Questions:**

**Q1:**
From the paper and figures I assume that the predictor outputs a series of temporally coarse frames, which are then predominantly used to refine the first prediction. The procedure is then repeated with this predicted and refined first prediction. To me, this means that $L>1$ predictions are made to advance the simulation by one frame. Is this the case? This should also be clarified in the paper.

**Q2:**
*Pfaff et al. (2021)* show that their method was stable across many steps. In Section 3.2 it is mentioned that *“[...] they are inferior in modeling long-term interacting systems that are continuous in nature and could suffer from severe error accumulation when making long-term predictions”*. Can you elaborate on this, especially regarding the inherent inferiority of GNNs?

**Q3:**
In Section 4.1: *“Instead of using inefficient iterative rollouts [...]”* - It is not entirely clear to me how iterative approaches are inherently inefficient at solving PDEs. To my understanding, even the method presented in the paper fundamentally relies on iterative temporal advancement of the solution.

**Q4:**
In Section 4.1: *“[...] thereby improving the capability to capture evolving patterns under potential noise”* - Can you clarify which evaluations from the paper hint at better performance under noise?

**Q5:**
In Section 4.1: *“While previous approaches often integrate neighborhood information into ODEs to model interacting dynamics, they typically fall short in explicitly capturing the evolving dynamics of edges”* - I do not fully understand this statement. Edges are not an inherent characteristic of the vector fields described by ODEs. Can you specify what you mean?

**Q6:**
In Section 4.2: The loss formulation $\mathcal{L}_{re}$ in equation (13) optimizes the refiner to match a ground truth $x^{t_1}$. The graph-ode predictor has, however, produced a series of future frames ($L$ to be precise). Is the refiner never trained to refine these steps?

**Details Of Ethics Concerns:**

As mentioned in the weakness **E2** above, this paper directly copies most results in Table 1 from *Cao et al. (2023)* and does not explicitly declare this. Only one hidden sentence in the reproducibility statement mentions this fact, while the rest of the paper and the appendix refers to the baselines results as coming from *“our implementation”*. I am not sure if this problem is sufficiently severe to fall within the definition of scientific misconduct, but I still wanted to explicitly raise this issue here as well, just in case. In my opinion this is problematic, as the results of MeshGraphNet reported by *Cao et al.* also clearly deviate from the original paper by *Pfaff et al.*, while using the same architecture on the same data sets.

---

> ### Author Response · Authors · 2023-11-22
> **Response to Reviewer vkK1 (I)**
>
> We thank the reviewer for the critiques that aim to improve the quality of all submissions. There are valuable points such as the confusion points, and there could be improvements to illustrate our method, that can help illustrate this paper better. Meanwhile, we also have to kindly clarify some misunderstandings:
> > “The method is evaluated against a series of baselines on steady-state physics scenarios.”
>
> Although we emphasize long-term stability. The loss is also calculated on all time snapshots, i.e. w.r.t the dynamic progress.
>
> > The reviewer kindly pointed out previous papers that use similar approaches or target similar problems as ours.
>
>
> First, we **claimed** our results are consistent with Cao et al. in **REPRODUCIBILITY STATEMENT** **in our first version**. Therefore, we didn't have **Ethics Concerns**.
> Second, as already noted by the reviewer the latter two papers, “DYffusion” and “PDERefiner” are too close to this submission to compare to. While the 1st paper was in the field of video interpolation. We also note that all three 3 methods use the diffusion model as their backbone and have only been tested on regular domains, in contrast to the irregular mesh of our case. While using the diffusion model for interpolation can be interesting, it’s orthogonal to our pipeline. Our focus is introducing this predicting-refining pipeline to the irregular mesh simulations.
> Other detailed replies:
>
>
> 1. “Given these limitations (on top of page 2)...”; we have refined our writing to clearly reflect that we are tackling the accumulated errors due to iterative rollouts.
> 2. “Our graph ODE model is flexible to generate…”; we have refined it to “Our model has the flexibility to generate the predictions at any steps ahead…”
> 3. “The model would be evaluated by calculating the prediction errors”; We use prediction errors w.r.t. the ground truth to evaluate the performance.
> 4. “However, they are inferior in modeling...”; We have deleted the discussion on previous methods and merged them into related works since this section is for methodology.
> 5. “To capture the hierarchy structure among meshes…”. Here is a typo, we have deleted that sentence.
> 6. “As a foundational step, it is crucial to…”; We have revised it into "As a preliminary step"
> 7. “To learn interacting dynamics…”; We have revised into "To learn the evolution between mesh points".
> 8. “Furthermore, we notice that data-driven…”; we have added the citations.
> 9. “Firstly, our FAIR…, Secondly, our approach… (appeared repeated in 4.1 and 4.3)”; We thank the reviewers for pointing out the confusion in our description. We have revised the presentation.
> 10. “Multi-task learning framework”; Our multi-task learning refers to the inference at the multiple time step forward into the future of the teacher model.
>
> 11. P2: Section 4 will be largely rewritten as mentioned in 9.
> 12. P3~P4: We thank you for pointing out and will illustrate them better
> 13. E1, E5: we use the full dataset of Pfaff et al. (2021). We did not evaluate any steady-state problems and would like to clarify to the reviewer that we only use a longer, coarser prediction in the future. We did not predict the steady-state in too-long future.
> 14. E2: We thank the reviewers for mentioning the soundness of our data source. The major problem we encountered is to strictly replicate the MeshGraphNet based on their open-source code. There exist multiple inconsistencies between their article and the code, one biggest issue we noticed is the noise added to the trajectory. While the article mentioned they use random-walk noise, in the code (https://github.com/google-deepmind/deepmind-research/blob/f5de0ede8430809180254ee957abf36ed62579ef/meshgraphnets/dataset.py#L75) they use normal distribution (without integrate on the time dimension to replicate random-walk effect). Another difference is that in the paper, the authors mentioned they drew a small box (without detailed dimensions) around the airfoil to only calculate the RMSE inside; however, this is not reflected in the code. We re-implemented based on the open-source TensorFlow code but found much worse results compared to the original one. There are multiple replications online while Cao et al. (2023) have the most comprehensive coverage on experiments that are conducted under the same, consistent setting, and they also provide a new dataset. That’s the reason why we adopted their work.

---

> ### Author Response · Authors · 2023-11-22
> **Response to Reviewer vkK1 (II)**
>
> 15. E3:
> - We visualized the composited velocities in Fig 3 and Fig 4., respectively.
> - We will re-print based on different timestamps to visualize our model’s capability on long-term predictions
> - Considering the soundness of the original MeshGraphNet, we would like to re-emphasize the point already made in 14.
> - Fig 14.a. Plot was mistakenly plotted with another component, hence we would like to fix that in the refined version.
> 16. E4:
> - We appreciate that you mentioned adding more clarity and would like to provide the following in the revised figure: the fixed L, and r when the other parameter is swiping.
> - We note that although increasing the time step is possible in other methods, that way they lose the detailed rolling out during these big timesteps. Also, using unrolled future states (i.e., the repeated current states) for the interpolation part in previous methods is no different than increasing the MLP layers. We also appreciate the reviewer for mentioning a few references, nonetheless, 1st they all need to include the traditional solvers in the loop, 2nd they work specifically for fluid dynamics applications. In contrast, our method does not need any domain knowledge and only relies on the training data.
> - Sure, we would love to revise the plot’s y-axis to log the scale.
> 17. Q1: Yes; your suggestion is accurate, we will make it clear in the revision
> 18. Q2: Considering the soundness of the original MeshGraphNet, we would like to re-emphasize the point already made in 14.
> 19. Q3: Claiming the iterative rolling out is inefficient was indeed a mistake, we will refine it better
> 20. Q4: Here we mean that one-step predictions would inevitably generate noisy predictions as the input during rollout iterations. Therefore, the long-term performance comparison can support this conclusion.
> 21. Q5: Here we intended to transit from NODE to graphODE. We will refine in our revision that “GraphODE is more suitable to explicitly capture the interaction on graph/between nodes via edges”
> 22. Q6: The L coarser future states are produced to be the input for the refiner, hence the refiner is still trained.
>
>
> In light of these responses, we hope we have addressed your concerns, and hope you will consider raising your score. If there are any additional notable points of concern that we have not yet addressed, please do not hesitate to share them, and we will promptly attend to those points.

---

> > ### Comment · Reviewer_vkK1 · 2023-11-22
> > **Reponse to the Rebuttal**
> >
> > I would like to thank the authors for their response, and I also looked at the revised version of the paper. The comments and changes address the comparatively minor presentation issues **P1** and **P4**, as well as some of the questions. Furthermore, judging if the substantial differences in Table 1 compared to the work from Pfaff et al. (2021) stem from issues in the MeshGraphNet source code or in the authors implementation of the method is hard to tell. This would most likely require a detailed reproducibility study of their paper. Nevertheless, such aspects should be ***clearly*** marked in the paper, in addition to the source of the values in Table 1. Skipping this crucial information, intentional or not, is a questionable research practice, reduces reproducibility, and could be considered plagiarism, thus raising ethical concerns.
> >
> > *However, the remaining fundamental issues with the evaluations and the corresponding presentation are not adequately addressed, neither in the comments nor in the revised version of the paper.*
> >
> > Most importantly, as this paper targets steady-state physics problems, there is no dynamic progress, i.e. no temporal physical change once the steady state is reached. The iterative simulation is just a computational tool due to lacking an analytical solution. As such, long-term stability for the investigated problems is meaningless, because no further computations are required beyond reaching a steady state. To show meaningful long-term stability, the investigation of unsteady problems is required. For fair evaluations on steady-state problems, all baselines should receive the same time steps (which should be as large as possible), as intermediate steps are not of interest for steady-state problems (since the physical behavior is steady, i.e. unchanging over time in the first place). Furthermore, comparisons to methods that directly predict the solution without the intermediate steps are highly necessary as they simply do not accumulate errors over a rollout, as mentioned in **E5**.
> >
> > While the authors mention that major parts of the paper will be adjusted or rewritten, these larger changes are beyond the scope of a conference rebuttal and should be thoroughly reviewed to ensure soundness and sufficient rigor. As a result, my original evaluation and thus the recommendation of strong reject for the current state of the paper remains.

---

> ### Author Response · Authors · 2023-11-23
> **Thanks for your response!**
>
> We are truly grateful for your response.  In the following response, we would like to address your major concern and provide additional clarification.
>
> > Skipping this crucial information (i.e., Pfaff et al. (2021)), intentional or not, is a questionable research practice, reduces reproducibility, and could be considered plagiarism, thus raising ethical concerns.
>
> Thanks for your comment. Actually, our paper didn't follow the settings in Pfaff et al. (2021) and just fully understood their settings to answer the concern. Moreover, we emphasized that we follow the setting in Cao et al. (2023), the new SOTA in this field. It seems that you **subjectively** think "we skip this" and "ethical concerns", which is **disappointing and disrespectful**. BTW, why do you think there is "plagiarism"?
>
> > This paper targets steady-state physics problems, there is no dynamic progress, i.e. no temporal physical change once the steady state is reached.
>
> Thanks for your comment. Actually, extensive literature [**Xu Han, Han Gao, Tobias Pfaff, Jian-Xun Wang, Li-Ping Liu**, Predicting Physics in Mesh-reduced Space with Temporal Attention, ICLR 22] studies long-term predictions for mesh-based simulations. **Are you saying these previous contributions in the field are meaningless?** Moreover, we have clearly stated that we did not predict the steady-state in the too-long future. In other word, no steady state is reached in our problem.
>
> Please let us know if you have further questions.

---

### Official Review · Reviewer_kYvB · 2023-10-31

**Soundness:** 3 good
**Presentation:** 3 good
**Contribution:** 3 good
**Rating:** 6
**Confidence:** 3

**Summary:**

The authors propose a method for mesh-based physical simulations focusing on improve the performance of long-term predictions. The proposed method first learn a graph ODE model for modelling coarse long-term predictions, subsequently enhancing the short-term predictions through interpolation. A continuous graph ODE model is utilized to integrate previous states in the progression of interacting node representations, which make it possible to capture long-term trajectories within a multi-task learning framework. The method can output long-term trajectories achieving considerable error reduction rate on benchmark datasets comparing to baselines.

**Strengths:**

- The idea of using future coarse long-term predictions to refine the short-term predictions is be sound and novel to me. Struggling to capture long-term dynamics is an important issue for the autoregressive style learning-based physical simulators due to the error accumulations over the iterative rollouts. The proposed method tackle this problem from a new perspective and has the potential to be further investigated.
- The experiments shown that the proposed method can obviously outperform the existing methods on the benchmark datasets.
- The paper is well written and organized with profound analysis.

**Weaknesses:**

- The current experiments provided in the paper are mainly in 2D. It is unclear whether the proposed method can exhibit similar performance when applied to larger scale 3D cases.
- There are no experiments for generalization ability for the proposed models. It is unclear how the trained models will perform given a test dataset generated by a variant (or same equation with different parameters) of the PDE which used to generate the training dataset.

**Questions:**

- A "multi-task learning framework" is mentioned multiple times in the paper for the first stage training. Does it indicate the encoder and decoder as multiple tasks?
- As the equations to solve in the experiments are generally PDEs, however the coarse long-term predictor i.e., Graph ODE is designed based on ODE. I am wondering are there any intuitions for how a ODE based neural solver can be applied on PDEs?
- Comparing to the existing autoregressive style methods, the refinement module in the proposed model is further trained on future states, which indicates the model get more chances to fit to the training data, I am wondering whether it may have overfitting issues? How does the model's generalization ability comparing to the existing autoregressive methods?

---

> ### Author Response · Authors · 2023-11-22
> **Response to Reviewer kYvB**
>
> We are truly grateful for the time you have taken to review our paper, your insightful comments and support. Your positive feedback is incredibly encouraging for us! In the following response, we would like to address your major concern and provide additional clarification.
>
> > Q1: The current experiments provided in the paper are mainly in 2D. It is unclear whether the proposed method can exhibit similar performance when applied to larger scale 3D cases.
>
> A1: Thanks for your comment. Actually, our experiment includes deformation plates and inflating fonts, which are 3D cases. Particularly, the inflating font contains 10s of thousands of nodes which are the largest among all cases. We have included a 3D visualization to show our performance for 3D mesh-based predictions.
>
>
>
> > Q2: There are no experiments for generalization ability for the proposed models. It is unclear how the trained models will perform given a test dataset generated by a variant (or the same equation with different parameters) of the PDE which used to generate the training dataset.
>
> A2: Thanks for your comment. We have examined the zero-shot generalization capability of our proposed FAIR, which tests the well-trained model in a different system without finetuning. In particular, we consider two systems, i.e., CylinderFlow and Airfoil, and transfer the model from one system to the other system. The compared results are shown below. From the results, we can find that although the performance decreases without finetuning, our FAIR always achieves better performance than the competing method, which verifies the high generalization capability of our FAIR.
>
>
>
> |  | Cylinder -> Cylinder $(\times 10^{-3})$ | Cylinder -> Airfoil $(\times 10^{-1})$ | Airfoil -> Airfoil $(\times 10^{-1})$| Airfoil -> Cylinder $(\times 10^{-3})$ |
> | - | - | - | -| - |
> | BSMS-GNN | 2.04 | 3.82 | 2.88 | 6.59 |
> | FAIR | 1.75 | 3.49 | 1.88 | 4.57 |
>
>
>
> > Q3: A "multi-task learning framework" is mentioned multiple times in the paper for the first stage training. Does it indicate the encoder and decoder as multiple tasks?
>
> A3. Thanks for your comment. Here, we generate predictions with different prediction lengths, each of which corresponds to a task. These tasks share the same encoder and decoder.
>
> > Q4: As the equations to solve in the experiments are generally PDEs, however the coarse long-term predictor i.e., Graph ODE is designed based on ODE. I am wondering are there any intuitions for how a ODE based neural solver can be applied on PDEs?
>
> A4. Thanks for your comment. Compared with ODE, PDE additionally includes both spatial derivates such as $u_x$ and $u_u$, which correspond to the relation between nodes and their neighbors. Therefore, combining neural ODEs with graph neural networks can simulate PDE solvers.
>
> > Q5: Comparing to the existing autoregressive style methods, the refinement module in the proposed model is further trained on future states, which indicates the model gets more chances to fit to the training data, I am wondering whether it may have overfitting issues? How does the model's generalization ability compare to the existing autoregressive methods?
>
> A5. Thanks for your comment. Since our foresight steps would generate a range of coarse predictions with potential noise, which would serve as the perturbation to the input for the refinement step. Therefore, with the noise as regularization, our FAIR wouldn't have a serious overfitting issue. The generalization capacity in A2 also validates our generalization capacity empirically.
>
>
> Thanks again for appreciating our work and for your constructive suggestions. Please let us know if you have further questions.

---

### Meta-Review · Area_Chair_pgho · 2023-12-06

**Metareview:**

This paper presents a two-stage, coarse to fine approach for predictions of physical systems with graph networks. It had received mixed reviews, with 2 reviewers providing borderline / slightly positive assessments, and 2 reviewers being very highly critical of the paper. The following issues were raised:

- restrictions to the 2D case (which the AC considered as minor issue)
- lack of experimental evaluation and generalization abilities,
- results being restricted to the steady state regime,
- issues with presentation and writing,
- generally, concerns that various claims of the paper were not backed up by results.

In particular reviewer vkk1 raised a long list of serious issues, among which probably the most severe were inconsistencies in the presentation of baseline results, questions on whether the results where taken or reimplemented, inconsistencies in the presented results vs. the results in the original papers.

The paper has been discussed between the authors and the most critical vkk1, who remained unconvinced and highly critical.

Having studied the reviewers answers, the AC sides with the critical reviewers, mostly on grounds of missing and unconvincing answers, and judges that the paper is not yet ready for publication.

**Justification For Why Not Higher Score:**

-

**Justification For Why Not Lower Score:**

-

---

### Decision · Program_Chairs · 2024-01-16

Reject